



# Annually resolved $\delta^2$H tree-ring chronology of the lignin methoxyl groups from Germany reflects averaged Western European surface air temperature changes

Anhäuser, Tobias[1*]; Sehls, Birgit[1]; Thomas, Werner[2]; Hartl, Claudia[3]; Greule, Markus[1];
Scholz, Denis[4], Esper, Jan[3] and Keppler, Frank[1,5*]

[1]Institute of Earth Sciences, Heidelberg University, Im Neuenheimer Feld 236, D-69120 Heidelberg, Germany
[2]Deutscher Wetterdienst, Meteorological Observatory station Hohenpeißenberg, Germany, Albin-Schwaiger-Weg 10, D-82383 Hohenpeissenberg, Germany
[3]Department of Geography, Johannes Gutenberg University of Mainz, Johann-Joachim-Becher-Weg 21, 55128 Mainz, Germany
[4]Institute of Geosciences, Johannes Gutenberg University of Mainz, Germany, Johann-Joachim-Becher-Weg 21, D-55128 Mainz, Germany
[5]Heidelberg Center for the Environment HCE, Heidelberg University, D-69120 Heidelberg, Germany

*Correspondence to: Tobias Anhäuser (tobias.anhaeuser@geow.uni-heidelberg.de) and Frank Keppler (frank.keppler@geow.uni-heidelberg.de)

**Abstract.** Stable hydrogen isotopes ratios of lignin methoxyl groups (expressed as $\delta^2$H$_{LM}$) of wood have been shown to reflect the climate-sensitive $\delta^2$H values of precipitation (expressed as $\delta^2$H$_{precip}$) modulated by a large uniform negative isotope fractionation. However, a detailed calibration study among temporal variabilities of $\delta^2$H$_{LM}$ in tree-ring series, site-specific $\delta^2$H$_{precip}$ and climate parameters has not been performed yet. Here, we present annually resolved $\delta^2$H$_{LM}$ values from nine tree-ring series (derived from four *Fagus sylvatica* L. trees) collected near stations of the Global Isotope Network of Isotopes in Precipitation (GNIP) and the Deutsche Wetterdienst (DWD) meteorological observatory at Hohenpeißenberg (southern Germany; ~48°N, 11°E). The measured nine $\delta^2$H$_{LM}$ tree-ring series (common period of overlap 1916-2015) show a strong coherency as indicated by highly significant ($p<0.001$) inter-series correlations (mean value Rbar = 0.52) and no indication of any known (stable isotope-specific) juvenile trend affecting the first decades of growth. When compared to local instrumental data, the combined $\delta^2$H$_{LM}$ chronology shows highest correlations with annually averaged data of $\delta^2$H$_{precip}$ as well as temperature particularly when using the year defined from previous September to current August (r = 0.73 and 0.56, respectively, $p<0.001$). However, the Hohenpeißenberg $\delta^2$H$_{LM}$ chronology shows enhanced correlations with land and sea surface air temperature for multiple (broadly combined) areas across Western Europe (r > 0.6, $p<0.1$). We subsequently established a linear regression model between averaged Western European surface air temperatures (range: 30°W–20°E, 35–60°N) and the $\delta^2$H$_{LM}$ chronology (r = 0.71, $p<0.001$). When comparing instrumental and reconstructed large-scale temperature anomalies from the year 1916 to 2015, an average absolute deviation in annual reconstructions of as low as 0.3 °C was found (n = 100). Therefore, $\delta^2$H$_{LM}$ values of mid-latitudinal tree-ring archives are considered suitable for large-scale mean annual temperature reconstructions and are therefore able to improve the paleoclimatic potential of Late Holocene tree-ring archives.



**Keywords**: compound-specific stable isotopes, paleoclimate proxy, tree-ring archive, stable water isotopes, temperature reconstructions, moisture source area

## 1. Introduction

High-resolution tree-ring series are valuable climate archives, particularly for the Late Holocene, fundamentally contributing to the understanding of past and current climate variability (Büntgen et al., 2011; Esper et al., 2002; Mann et al., 1998). Dendrochronological climate proxies are commonly used to reconstruct local temperature variability derived from plant physiological parameters, such as tree-ring width (TRW) or maximum late wood density (MXD) from trees growing at or close to the altitudinal or latitudinal treeline. Alternative proxies potentially suitable for climate reconstructions of non-treeline regions are tree-ring stable hydrogen and oxygen isotope ratios (expressed as $\delta^2H$ and $\delta^{18}O$ values) of stem wood as they reflect partly stable water isotopes of the local precipitation ($\delta^2H_{precip}$ and $\delta^{18}O_{precip}$) (Hartl-Meier et al., 2014; Liu et al., 2015; McCarroll and Loader, 2004; Pauly et al., 2018; Sternberg, 2009; Treydte et al., 2006). Both $\delta^2H$ and $\delta^{18}O$ values are sensitive tracers of hydroclimate as they integrate a combination of (i) meteorological conditions in the moisture source area, (ii) changes during the meridional atmospheric vapor transport (particularly water volume loss, but also potential mixing of different air masses), and (iii) local temperature (Araguas-Araguas et al., 2000; Dansgaard, 1964; Rozanski et al., 1993). Hence, reconstructions of $\delta^2H_{precip}$ and $\delta^{18}O_{precip}$ values are not only able to broaden the applicability of tree-ring archives for non-treeline regions, but may also complement commonly used tree-ring proxies (reflecting local climate changes) with an additional stable isotope proxy (reflecting local and large-scale hydroclimatic changes).

The compound of choice for paleoclimate investigation is commonly cellulose since their stable oxygen and hydrogen isotopes signatures are derived from the climate-sensitive stable water isotopes. However, cellulose incorporates hydrogen and oxygen atoms partly from leaf-water that experienced an evaporative $^2H$ and $^{18}O$ enrichment prior to biosynthesis potentially complicating the reconstruction of the initial source water $\delta^2H$ and $\delta^{18}O$ values (Flanagan et al., 1991; Roden et al., 2000). Furthermore, the extraction procedures are time consuming. Thus, studies are commonly forced to pool samples or build chronologies using only low or even no replication hampering the statistical evaluation of the accompanied $\delta^2H$ and $\delta^{18}O$ variability in high resolution tree-ring series (McCarroll and Loader, 2004).

Another wood component that has been increasingly employed for the investigation of $\delta^2H$ values in tree-ring series are lignin methoxyl groups (expressed as $\delta^2H_{LM}$ values). Methoxyl groups (R-OCH$_3$) in wood are predominantly ether bonded in lignin including three non-exchangeable hydrogen atoms that do not exchange with other hydrogen containing organic compounds or surrounding water. Hence, contrary to the cellulose-derived $\delta^2H$ values, $\delta^2H_{LM}$ values are not influenced by an evaporative $^2H$ enrichment of leaf water since the wood component lignin is formed within the xylem tissue. Furthermore, $\delta^2H_{LM}$ values of wood can readily be measured as methyl iodide (CH$_3$I) upon treatment with hydroidic acid providing a fast and straightforward extraction method. Earlier studies mainly evaluated and quantified the spatial $\delta^2H_{precip}$-$\delta^2H_{LM}$ relationship using numerous sampling sites (Anhäuser et al., 2017a; Feakins et al., 2013; Keppler et al., 2007), but also investigated site-specific $\delta^2H_{LM}$ variabilities between trees at single sites (Anhäuser et al., 2017b). These studies suggest that $\delta^2H_{LM}$ values reflect $\delta^2H$ values of the tree's source water ($\delta^2H_{sw}$ values) as significant relationships between



$\delta^2H_{LM}$ and mean annual $\delta^2H_{precip}$ values could be observed (implying that $\delta^2H_{sw}$ values reflects an annual integral
of the site-specific $\delta^2H_{precip}$ value). As $\delta^2H_{sw}$ values were not measured in most of these studies the estimated
isotope fractionation was calculated between $\delta^2H_{precip}$ and $\delta^2H_{LM}$ (commonly expressed as the apparent isotope
fractionation $\varepsilon_{app}$) and is considered to reflect primarily the biosynthetic isotope fractionation ($\varepsilon_{bio}$) during lignin
methoxyl biosynthesis.

Calculations of $\varepsilon_{app}$ showed a broad agreement between coniferous and deciduous tree-species (n = 13) with $\varepsilon_{app}$
values of –213 ± 17 ‰ except for *Picea abies* (–237 ± 19 ‰; Anhäuser et al., 2017a). At a single site, $\delta^2H_{LM}$
values showed maximum differences of ≤ 28 ‰ between five trees (within a circumference of 200 m). Hence,
the accompanied variability of $\varepsilon_{app}$ (in the range of ± 17 to 19 ‰) was mainly assigned to inter-tree variability
and defines (at this stage of research) the accuracy to reconstruct 'absolute' $\delta^2H_{precip}$ values using $\delta^2H_{LM}$ values.
Reconstructions of 'absolute' $\delta^2H_{precip}$ values have therefore been considered suitable for when performed on
wood samples covering anticipated large $\delta^2H_{precip}$ changes (exceeding the accompanied $\varepsilon_{app}$ variability of ± 17 to
19 ‰) as could recently been shown for deep-time Eocene fossil wood specimen (Anhäuser et al., 2018).

Anhäuser et al. (2017a) suggested that relative temporal $\delta^2H_{LM}$ changes, measured from multiple tree-ring series,
may reflect relative temporal $\delta^2H_{precip}$ changes much more accurately since the characterized main noise of $\varepsilon_{app}$
(inter-tree variability) is minimized. Earlier studies conducted preliminary investigations of inter-annually
resolved $\delta^2H_{LM}$ tree-ring series (Gori et al., 2013; Mischel et al., 2015; Riechelmann et al., 2017). However, a
detailed evaluation of the temporal $\delta^2H_{LM}$-$\delta^2H_{precip}$ relationship was not possible either due to lacking site-
specific instrumental $\delta^2H_{precip}$ data (Gori et al., 2013; Mischel et al., 2015; Riechelmann et al., 2017) or
potentially insufficient tree-ring series length of two decades (Mischel et al., 2015). Furthermore, Gori et al.
(2013) and Riechelmann et al. (2017) used trees growing predominantly on thin soils and steep slopes as
sampling was carried out in mid- to high-altitudinal Alpine regions (~900, ~1300, ~1500 and ~1900 m.a.s.l.). At
such sites, tree source water is considered prone to $\delta^2H$ changes due to a low residence time and capacity of soil
water leading to seasonally biased soil water accumulation.

To evaluate in detail the significance of $\delta^2H_{LM}$ to reflect $\delta^2H_{precip}$ and its potential for climate reconstructions,
this study investigates tree-ring $\delta^2H_{LM}$ variability in the vicinity (< 2 km) of stations of the Global Network of
Isotopes in Precipitation (GNIP) and the Deutsche Wetterdienst (DWD) located at Hohenpeißenberg (Germany).
Nine cores from four *Fagus sylvatica* L. specimens were used to perform inter-annually resolved $\delta^2H_{LM}$
measurements with a common period of overlap 1916-2015. The data are used to assess the co-variance among
radii and different trees and to evaluate potential non-climatic, age-related influences. Subsequently, the $\delta^2H_{LM}$
tree-ring series are compared to site-specific $\delta^2H_{precip}$ reference data as well as local and large-scale climate data.

## 2. Study site and local climate

The study was conducted around the Hoher Peißenberg Mountain near the Hohenpeißenberg municipality in the
northern Alpine foothills in Germany (47° 48' N, 11° 01' E; ~800 m.a.s.l.; Fig. 1). Mean annual temperatures
(MAT) during the 1916-2015 period ranged from 4.8 to 8.5 °C with a mean value of 6.6 °C and monthly
temperatures varying between –1.4 °C (January) and 15.2 °C (July) (Fig. 2a,b). For the same time period, total
annual precipitation ranged between 780 and 1570 mm averaging 1160 mm, whereby half of the total annual
precipitation falls between May and August. Monthly resolved $\delta^2H_{precip}$ data are available from 1971-2008 from
the nearby GNIP station (Fig. 2c, d). Weighted mean annual $\delta^2H_{precip}$ values show an average value of –75 ‰



with a seasonal range from −107 (February) to −52 ‰ (August). Long-term climate trends of the 1916-2015 period at Hohenpeißenberg show that MATs increased significantly by 0.12 °C per decade (r = 0.56, p < 0.001),

whereby neither significant changes in annual nor seasonal precipitation are observed (considering meteorological seasons). For the 1971-2008 period, weighted mean annual $\delta^2H_{precip}$ values correlate highly significantly with MAT (r = 0.52, p < 0.001), but show neither a significant correlation with annual precipitation (r = −0.05, p = 0.62) nor with seasonal precipitation (r ranging from −0.32 to 0.2, p > 0.05).

## 3. Material and methods

### 3.1 Tree-ring material

Twenty *F. sylvatica* trees were sampled in spring and autumn 2016 at Hohenpeißenberg. We took two to three cores per tree at breast height (~1.2 m above ground) with a 5-mm increment borer yielding a total number of 52 cores (core lengths ranged from 400 to 600 mm). The four *Fagus sylvatica* trees (referred to as F1, F2, F3 and F4; cf. Fig. 1) selected for $\delta^2H_{LM}$ measurements were either located at moderate east-facing slopes (< 10%), such

as F1 and F2, or in flat terrain, such as F3 and F4.

After collection, increment cores were cut perpendicular to the wood fibers using a core-microtome to ease ring identification (Gärtner and Nievergelt, 2010). Tree-ring width (TRW) was measured for all cores with an accuracy of 0.01 mm using a LINTAB measuring table equipped with TSAP-Win Software (both Rinntech, Heidelberg, Germany). Cross-dating accuracy was assessed visually and statistically using the program

COFECHA (Holmes, 1983). For $\delta^2H_{LM}$ analysis, four trees (F1 to F4) with no obvious disturbance influence, i.e. growth release or suppression, were selected. To better assess intra-tree variability we choose three cores of F1 and two cores of F2, F3 and F4 (termed F1.1, F1.2, F1.3 and so on, Table 1). The longest tree-ring series of the four *Fagus sylvatica* trees date back to 1905 (F1.2), 1844 (F2.2), 1890 (F3.1) and 1896 (F4.1) and were used to determine the 'pith offset' (number of missing innermost rings). The corresponding shortest tree-ring series for

each tree dated back to 1912 (both F1.1 and F1.3), 1858 (F2.1), 1916 (F3.2) and 1916 (F4.2). Consequently, each *Fagus sylvatica* tree provides a minimum of two radii that covered the 1916-2015 common period whereby two radii older than 1916 are particularly provided by F2 (extending back to 1844 and 1858). Over the full periods, tree rings were carefully dissected at the tree ring border (latewood/earlywood transition) with a scalpel under a magnifier to subsequently allow annually resolved $\delta^2H_{LM}$ measurements. All dissected tree-ring samples

yielded a minimum weight of 1 mg.

### 3.2 Instrumentation and analytical uncertainty of $\delta^2H_{LM}$ analysis

The $\delta^2H_{LM}$ value of wood can be measured as $CH_3I$ released upon treatment of the samples with hydroiodic acid (HI) by the method described by Greule et al. (2008). Briefly, HI (0.25 ml; puriss. p.a., 55-60%, not stabilized,

purchased from Sigma-Aldrich, Seelze, Germany, or Gillingham, UK, respectively) was added to the annually dissected tree-rings (1 to 10 mg, not homogenized) in a crimp glass vial (1.5 ml; IVA Analysentechnik, Meerbusch, Germany). The vials were sealed with crimp caps containing PTFE lined butyl rubber septa (thickness 0.9mm) and incubated for 30 minutes at 130°C. After heating, the samples were subsequently allowed to equilibrate at room temperature (22 ± 0.5 °C, air conditioned room) for at least 30 minutes before an aliquot of



the headspace (10-90 μl) was collected and directly injected into the analytical system using a gastight syringe (100 μl, SGE Analytical Science).

$\delta^2$H values of the $CH_3I$ formed from the methoxyl groups in the wood samples ($\delta^2H_{LM}$) were measured using an HP 6890N gas chromatograph (Agilent, Santa Clara, USA) equipped with an auto sampler A200S (CTC Analytics, Zwingen, Switzerland), coupled to a Delta$^{PLUS}$XL isotope ratio mass spectrometer (ThermoQuest

Finnigan, Bremen, Germany) via a thermo conversion reactor [ceramic tube ($Al_2O_3$), length 320 mm, 0.5 mm i.d., reactor temperature 1450 °C] and a GC Combustion III Interface (ThermoQuest Finnigan, Bremen, Germany). The gas chromatograph (GC) was fitted with a Zebron ZB-5MS capillary column (Phenomenex, Torrance, USA) (30m x 0.25mm i.d., $d_f$ 1μm), and the following conditions were employed: split injection (4:1), initial oven temperature at 30 °C for 3.8 min, ramp at 30 °C/min to 100 °C. Helium was used as carrier gas at a

constant flow rate of 0.6 mL/min.

A tank of high purity hydrogen gas ($H_2$, hydrogen 5.0, Linde, Höllriegelskreuth, Germany) with a $\delta^2$H value ranging from –195 to –225 ‰ (vs. V-SMOW, range provided by the supplier) was used as the working reference gas. The $H_3^+$ factor, determined daily during the measurement period, was between 1.9 and 2.4 ppm/nA.

All $\delta^2H_{LM}$ values were normalized by a two-point linear calibration (Paul et al., 2007) using $\delta^2$H values of two

$CH_3I$ working standards relative to V-SMOW. The $\delta^2$H values of the $CH_3I$ working standards were calibrated against international reference substances (VSMOW2 [$\delta^2H_{VSMOW} = 0.0 \pm 0.3$ ‰] and SLAP2 [$\delta^2H_{VSMOW} = -427.5 \pm 0.3$ ‰]) using TC/EA-IRMS (elemental analyser-isotopic ratio mass spectrometer, IsoLab, Max Planck Institute for Biogeochemistry, Jena, Germany). The calibrated $\delta^2$H values in ‰ versus V-SMOW for the two $CH_3I$ working standards were $-173.0 \pm 1.5$ ‰ (n=9, 1σ) and $-66.2$ ‰ $\pm 1.2$ ‰ (n=8, 1σ). All samples were

measured in triplicate followed by consecutive injections of both working standards. Standard deviations (1σ) of the triplicate measurements averaged $\pm$ 1.4 ‰ (ranging from $< \pm 0.1$ to 13.9 ‰). Additional uncertainty is introduced by the 'external precision' (also referred as the chemical replication uncertainty). To estimate this uncertainty, we cut tree rings (n = 68) parallel to growth direction to obtain three tree-ring subsamples of the same cambial age (each comprising early and late wood). These subsamples enabled three separate HI treatments

and subsequent $\delta^2H_{LM}$ analysis. The standard deviation of the three subsamples (n = 68) averaged $\pm$ 2.7 ‰ (1σ) and was used as estimator for the 'external precision'. Using Gaussian error propagation, the overall analytical uncertainty of the $\delta^2H_{LM}$ determination averaged $\pm$ 3.6 ‰ (ranging from $\pm$ 3.3 to $\pm$ 14.4 ‰ with n = 1086).

### 3.3 $\delta^2H_{LM}$ covariance assessment and calibration

Coherency among the nine $\delta^2H_{LM}$ tree-ring series was assessed over the 1916-2015 common period using

Pearson's correlation coefficient (r) as well as the inter-series correlation (Rbar). To identify potential non-climatic trends of $\delta^2H_{LM}$ values of juvenile tree-rings, we compared linear trends among differently old trees over a common time period. First year autocorrelation (lag-1) has been calculated to determine the influence of the previous year on $\delta^2H_{LM}$.

Since tree-ring $\delta^2H_{LM}$ is primarily derived from $\delta^2H_{precip}$, we firstly calculated Pearson's correlation coefficients

from 1971-2008 between these data using a variety of monthly, seasonal and annual mean values to evaluate the tree´s source water. Moreover, tree source water can - to some extent - involve the precipitation of the previous year (Ehleringer and Dawson, 1992; Feng and Epstein, 1995; Tang et al., 2000). Hence, we also included $\delta^2H_{precip}$ signatures of months of the preceding fall as well as 'shifted' annual values, such as previous September



to current August. Whenever $\delta^2H_{precip}$ values were averaged, individual $\delta^2H_{precip}$ values were weighted by precipitation amount to obtain representative mean values. Subsequently, correlation coefficients between $\delta^2H_{LM}$ tree-ring series and local climate parameters of Hohenpeißenberg were assessed for the common period of overlap (1916-2015) employing data of the nearby DWD weather station. To assess large-scale temperature influences on the Hohenpeißenberg tree-rings, spatial correlations between $\delta^2H_{LM}$ and the HadCRUT4.6 ensemble data (Morice et al., 2012) were conducted over the 1916-2015 period. Due to the prevailing westerlies

across Europe, we focused on areas mainly west of Hohenpeißenberg (~48 °N, 11 °E). Thus, spatial correlations with land and sea surface air temperature (CRU TS 4.01 and HadSST1, respectively) were conducted from North Africa to southern Scandinavia and from the Eastern North Atlantic to mid-Europe (range: 35 – 60°N and 30°W – 20°E) employing the KNMI climate explorer at 1° x 1° resolution (Royal Netherlands Meteorological Institute; http://climexp.knmi.nl). The reconstruction skill for $\delta^2H_{LM}$ tree-ring series and climate parameters was assessed

by the Durbin-Watson statistics (DW) testing for lag-1 autocorrelation in the linear model residuals.

### 4. Results and discussion

#### 4.1 Intra- and inter-tree variability of the $\delta^2H_{LM}$ series

For the common period of overlap (1916-2015), maximum and minimum $\delta^2H_{LM}$ values of the nine tree-ring series ranged from –243 to –224 ‰ and –295 to –270 ‰, respectively, yielding maximum $\delta^2H_{LM}$ differences

within a core ranging between 41 and 66 ‰ (Table 2). Mean $\delta^2H_{LM}$ values ranged from –250 ± 8 to –271 ± 11 (1σ standard deviation), whereby the magnitude of this range primarily results from $\delta^2H_{LM}$ differences among trees rather than between radii. Hence, in agreement with earlier findings (Anhäuser et al., 2017b; Riechelmann et al., 2017), $\delta^2H_{LM}$ values can differ regarding 'absolute' values particularly between trees by up to 21 ‰. Anhäuser et al. (2017b) suggested that about half of the inter-tree variability results from site-specific $\delta^2H$

differences of the tree source water (i.e., $\delta^2H$ differences may already exist prior to water uptake), highlighting, however, that deviations in the biosynthetic isotope fractionation cannot be excluded.

Employing the full length of each of the nine tree-ring series, highly significant correlations (p < 0.001) can be noted among the $\delta^2H_{LM}$ series, whereby somewhat higher r values are observed within trees (ranging from 0.49 to 0.73) compared to inter-trees (ranging from 0.37 to 0.66; Table 3). Over the common period 1916-2015, inter-

series correlation (Rbar) yielded 0.52. Hence, even though $\delta^2H_{LM}$ values can differ regarding 'absolute' values particularly between trees by up to 21 ‰, the common variability indicates that these 'absolute' $\delta^2H_{LM}$ differences remain predominantly constant over time. Lag-1 is high and ranges between 0.40 and 0.78 indicating strong lag effects inherent to all $\delta^2H_{LM}$ series. These autocorrelations may reflect an autocorrelation adopted from the $\delta^2H_{precip}$ values. An equivalent autocorrelation for the $\delta^2H_{precip}$ over the 1971-2008 period cannot be

observed (r = 0.22, p > 0.1). However, a lack of autocorrelation in the $\delta^2H_{precip}$ data excludes not an absence of autocorrelation for $\delta^2H_{sw}$ values. Whereas $\delta^2H_{precip}$ values are individually measured of monthly collected precipitation, the source water $\delta^2H$ value reflects an integral of multiple precipitation events or even seasons reflecting a water body with a progressively changing $\delta^2H$ value. Thus, it seems (hydrologically) reasonable to assume an autocorrelation inherent to $\delta^2H_{sw}$ which is transferred to the $\delta^2H_{LM}$ tree-ring series. Nonetheless, the

autocorrelations observed for the $\delta^2H_{LM}$ tree-ring series may also point to some hydrogen 'storage effect' such as the incorporation of remobilized hydrogen during the lignin methoxyl groups biosynthesis.





To emphasize the coherent long-term $\delta^2H_{LM}$ variability of the four *Fagus sylvatica* trees, we calculated for each series the annual $\delta^2H_{LM}$ deviation to its mean value of the 1961-1990 period and averaged the resulting index $\delta^2H_{LM}$ series for each tree to produce four mean $\delta^2H_{LM}$ series (Fig. 3). For each mean series, both the short-term

(inter-annual) and long-term (multi-decadal) changes in $\delta^2H_{LM}$ values are comparable in magnitude with up to ~20 ‰. The long-term $\delta^2H_{LM}$ trends can broadly be grouped into four phases including *(1)* a decline of ~20 ‰ from 1858-1916 (covered only by F2), *(2)* a ~15 ‰ increase from 1916-1956 retained in all tees, *(3)* a decrease of ~10 ‰ until 1981, and *(4)* an increase by ~25 ‰ during the most recent decades. The most recent increase in $\delta^2H_{LM}$ values is most pronounced in F4.

## 4.2 Assessment of tree age related influences on $\delta^2H_{LM}$ values and mean chronology development

$\delta^2H_{LM}$ values may additionally be affected by non-climatic influences related to age or tree height. To test for this, we compared linear fits among the four mean $\delta^2H_{LM}$ series using the 1916-1956 period (above described as phase two) as here each series shows a significant positive linear trend sharing also a common positive $\delta^2H_{LM}$ excursion at the end of the period (1956; Fig. 3b). Moreover, during this 51 years period, the four *Fagus*

*sylvatica* trees reflect different age phases including a juvenile phase of F1 (cambial age 12-52), intermediate phases of F4 and F3 (cambial age 27-67 and 34-74, respectively) as well as a mature phase of F2 (cambial age 81-121). This allows a statistical assessment of differences among $\delta^2H_{LM}$ trends and tree age (cf. Table 4). The slopes of the linear regressions over the period 1916-1956 become steeper with decreasing tree age phase ($0.24 \pm 0.18$, $0.34 \pm 0.14$, $0.31 \pm 0.17$ and $0.49 \pm 0.18$ ‰ per year; Table 4; Fig. 4). However, when conducting a two-

tailed t-test for slope differences, only the linear fits of juvenile F1 data and the mature F2 data are statistically different at $p < 0.1$. Hence, the elevated slope of F1 compared to F2 suggests a juvenile effect of F1 associated with increasing $\delta^2H_{LM}$ values (0.25 ‰ per year). Even though statistically significant, the magnitude of this potential 'juvenile trend' falls notably below the inter-annual $\delta^2H_{LM}$ variability of up to 20 ‰ (Fig. 3) as well as the average analytical overall uncertainty of $\pm 3.6$ ‰ and, thus, should have a negligible effect on the general

coherency. A visibly stronger discrepancy in $\delta^2H_{LM}$ trends among the different trees can be observed for F4 at the end of our chronology (Fig. 3b). When comparing this discrepancy to F3, having a similar age as F4 (7 years apart; Table 1) suggests an influence not related to tree age. Consequently, deviations in the temporal $\delta^2H_{LM}$ trends seem not primarily associated with juvenile growth phases or tree age in general but may also be associated by temporal disturbances of $\delta^2H_{sw}$ among the four tree sampling sites, for instance, induced by

changes in the source water accumulation. A better understanding of such occasional temporal trend deviations in $\delta^2H_{LM}$ may improve sampling strategies. Trees with potential temporal disturbances in the $\delta^2H_{sw}$-$\delta^2H_{LM}$ relationship may justifiably be excluded when producing a mean $\delta^2H_{LM}$ chronology subsequently leading to an improved overall coherency.

For cellulose-derived $\delta^2H$ and $\delta^{18}O$ tree-ring series, there are commonly two mechanisms considered to induce

juvenile trends. First, arid study sites with high evaporation rates may lead to an evaporative enrichment of heavier stable isotopes in source waters of upper soil layers (Allison, 1982; Hsieh et al., 1998; Kanner et al., 2014). In presence of such a gradient in stable water isotopes in soils, trees would use consecutively less $^2H$ enriched soil waters over time as their root system progressively accesses deeper soil layers. Such a process would add a decreasing component to tree-ring $\delta^2H$ and $\delta^{18}O$ values until the root system is fully developed as

has been suggested for a cellulose-derived $\delta^{18}O$ series from the Mediterranean (Corsica, France; Szymczak et al., 2012). Another mechanism for juvenile trends may be induced by a gradient of relative humidity under the forest



canopy. A decrease in relative humidity with height has been observed in temperate forests (Eliáš et al., 1989; Zweifel et al., 2002), which would intensify $^{18}O$ enrichment as leaf water evaporation rates increase with tree height (Labuhn et al., 2014). Hence, with increasing tree height (or age), leaf water would become gradually

more enriched in heavy stable water isotopes. Contrary to the first mechanism, such a process would lead to increasing $\delta^2H$ and $\delta^{18}O$ values in juvenile tree-rings. However, for $\delta^2H_{LM}$ values, we consider only the first (soil water-related) mechanism as influential because leaf water-derived hydrogen is not involved in lignin methoxyl group's biosynthesis (Boerjan et al., 2003). Thus, as the juvenile $\delta^2H_{LM}$ values of F1 presented in this study show a minor mean trend deviation (associated with increasing $\delta^2H$ values), this indicates no impact of soil water-

related juvenile effects, which would be associated with decreasing $\delta^2H$ values. This may either point to a weak or nonexistent gradient of soil water $\delta^2H$ values at Hohenpeißenberg or to a quickly developed root system of *Fagus sylvatica* trees within the first decade of growth. Nonetheless, future studies should take into account the potential influence of soil water evaporation rates on juvenile $\delta^2H_{LM}$ values, in particular for more arid study sites than Hohenpeißenberg. Furthermore, the most pronounced discrepancy in the temporal $\delta^2H_{LM}$ trend noted

for F4, which is presumably not age-related, highlights to investigate other non-climatic influences as well. A plausible mechanism would be a changing relationship between $\delta^2H_{precip}$ and $\delta^2H$ of the source water over time in the nearby tree environment, such as changing soil and hydrological properties (leading to different source water accumulations).

We developed a chronology (without conducting any detrending or correction method) for the common period

1916-2015 by calculating for each of the nine tree-ring series the annual $\delta^2H_{LM}$ deviation to the mean of the 1961-1990 period. Subsequently, by integrating all nine $\delta^2H_{LM}$ series, a mean chronology was calculated with 95% confidence intervals to account for the interval variability (Fig. 5).

### 4.3 Relationship between $\delta^2H_{LM}$ chronology and $\delta^2H_{precip}$ as well as local and large-scale climate

The results clearly show that our mean $\delta^2H_{LM}$ chronology correlates best with weighted mean annual $\delta^2H_{precip}$

signatures using time periods such as January until December with r = 0.66 (p < 0.001; Fig. 6a) or previous September to current August with r = 0.73 (p < 0.001) when compared to seasonal and monthly $\delta^2H_{precip}$ signatures. The increased r value for the 'shifted' mean annual $\delta^2H_{precip}$ values agrees with the timing of early and late wood formation (growth season) and the source water accumulation. As tree-ring growth ceases in fall, precipitation falling after the latest late wood formation may only contribute to the tree source water of the

following year. Hence, in agreement with considerations in annual source water accumulation and tree-ring growth, the $\delta^2H_{LM}$ chronology is considered to reflect primarily mean annual $\delta^2H_{precip}$ values (previous September to current August) for the 1971-2008 period of data overlap. For the remaining years of the full 1916-2015 period, care has to be taken when the relative contribution of seasonal precipitation to the annual amount changes. In that case, $\delta^2H_{LM}$ would reflect a more seasonally pronounced $\delta^2H_{precip}$ signature. However, even

though this induces some noise into the $\delta^2H_{precip}$-$\delta^2H_{LM}$ relationship we consider this influence as negligible for Hohenpeißenberg since neither annual nor seasonal precipitation amount changes significantly over 1916-2015 (cf. Sect. 2).

When comparing the $\delta^2H_{LM}$ chronology with temperature data of the nearby DWD station (covering the full period of 1916 - 2015), highly significant correlations (p < 0.001) are found for numerous months (previous

October, January, April, June, August) with r values ranging from 0.26 to 0.39 and for all seasons except for fall of the current year with r values ranging from 0.29 to 0.41 (Fig. 6b). Similar as observed for $\delta^2H_{precip}$, highest



correlation coefficients can, however, be noted for annually averaged temperatures with r = 0.53 (year defined from January to December) and r = 0.56 (year defined from previous September to August). Precipitation amount does not show any highly significant correlation to the $\delta^2H_{LM}$ chronology (Fig. 6b).

As indication was found that the $\delta^2H_{LM}$ chronology reflects best annually averaged $\delta^2H_{precip}$ values considering the year prior to the latest late wood formation, spatial correlations have been accordingly estimated between the mean $\delta^2H_{LM}$ chronology from Hohenpeißenberg (Germany) and mean annual surface air temperatures of previous September to current August. The results show significance across Western Europe (r ranging from 0.2 to >0.6; Fig. 7a). Highest correlations (with r > 0.6) are found for both land (northern Mediterranean) and sea

areas (western Mediterranean Sea, northwest North Sea, Gulf of Biscaya and parts of the eastern North Atlantic). This shows that the $\delta^2H_{LM}$ chronology exhibits higher correlations with temperature anomalies of multiple remote areas than with local temperature anomalies at Hohenpeißenberg (r = 0.56; Fig. 5b). To test the temporal robustness of the relationship between the $\delta^2H_{LM}$ chronology and Western European surface air temperatures, spatial correlations have also been determined for the 50-year periods 1916-1965 and 1966-2015 (cf. Fig. 7b and

C). Similarly as observed for the full period, the Hohenpeißenberg $\delta^2H_{LM}$ chronology and surface air temperature anomalies show significant correlations for both periods for large areas across western and central Europe. However, areas of highest correlations expand eastwards and increase in magnitude with time (cf. Fig. 7b and c). This leads particularly to an enlarged coverage of areas of high correlation for European land masses.

Besides changes in local temperature, $\delta^2H_{precip}$ at the site of precipitation is controlled by a number of large-scale

and remote hydro-climatic influences such as changing meteorological conditions in the moisture source area and meridional atmospheric transport particularly a modified water volume loss, but also potential mixing with different air masses (Araguas-Araguas et al., 2000; Dansgaard, 1964; Rozanski et al., 1993). The main moisture source of precipitation of the northern Alpine foothills (including the Hohenpeißenberg study site) has been determined by Lagrangian moisture source diagnostics for the time period 1995-2002 (Sodemann and Zubler,

2010). This study concludes that the majority of annual northern Alpine precipitation originates from multiple sources including the eastern North Atlantic (~29 %), the Western Mediterranean Sea (~16%), the North and Baltic Sea (~13 %) and continental Europe (~21 %). Consequently, even though our $\delta^2H_{LM}$ chronology shows highly significant correlations with large-scale and remote annual temperature changes, the involved stable water isotope mechanisms on Hohenpeißenberg $\delta^2H_{precip}$ values are more complex considering the multiple moisture

source areas of northern Alpine precipitation and the associated numerous hydro-climatic influences therein. Interestingly is, however, that northern Alpine moisture source areas overlap broadly with areas where surface air temperature changes show the highest correlations with the Hohenpeißenberg $\delta^2H_{LM}$ chronology (r between 0.5 and >0.6; Fig. 7a). The overlap of both these areas may suggest that temperature changes in the (remote) moisture source areas are an important control of the site-specific $\delta^2H_{precip}$ variance at Hohenpeißenberg.

Furthermore, applying this finding to the observed eastward expansion of areas of highest correlation from 1916-1965 to 1966-2015 may point to an eastward expansion of moisture source areas of northern Alpine precipitation during the 20th century (from the European shelf to Western and Central Europe).

### 4.3 Reconstructions of averaged western European surface air temperature changes

Even though areas of highest correlations between surface air temperatures and the $\delta^2H_{LM}$ chronology expand

eastwards with time, those areas are connected across Western Europe and overlap largely over time (Fig. 7). Hence, the $\delta^2H_{LM}$ chronology may be suitable to reconstruct average surface air temperatures of this large-scale



area (30 °W - 20 °E; 35 - 60 °N; area displayed in Fig. 7). To test this we developed a linear regression model
between the annually averaged (previous September to August) Western European surface air temperature
(WESAT) anomaly and the $\delta^2H_{LM}$ chronology yielding r = 0.71 (p < 0.001, DW = 1.6). When expressing the
linear regression model, the following equation is obtained:

**Eq. (1)**: $\Delta\text{Temperature}_{\text{WESAT}}$ [°C] ≈ ($\Delta\delta^2H_{LM}$ [‰] + 0.98 ) / 15.88 [‰ / °C]

We subsequently compared the reconstructed and instrumental WESAT anomalies at annual resolution (Fig. 8).
To account for errors induced by the internal $\delta^2H_{LM}$ variability of the nine tree-ring series, we converted the
annually estimated 95% confidence interval (error bars in Fig. 5) into temperature using Eq. (1). As an additional
error assessment, we calculated the differences of the reconstructed and instrumental WESAT anomalies that
result at annual resolution and with a moving average of 5 (Fig. 8b). Differences between annually reconstructed
and instrumental WESAT anomalies differ in the range of Δ -0.9 to Δ 1.3 °C with an average absolute deviation
of 0.3 °C (n = 100) (Fig. 8b). Hence, for the broad majority of annual reconstructions (n = 84), absolute
deviations are < 0.5 °C, whereby the remaining absolute deviations in annual reconstructions are either between
0.5 and 1.0 (n = 14) and highest in 1956 and 1993. For the smoothed data (moving average of 5 data points), the
agreement between reconstructed and instrumental WESAT anomalies can particularly be observed for the main
part of our chronology (1937 - 2007), where absolute differences are entirely < 0.3 °C (Fig. 8b). Prior and after
this period, elevated absolute differences between the reconstructed and instrumental WESAT anomalies of up to
0.5 °C can occur for several consecutive years. These periods of elevated absolute differences broadly coincide
with the discrepancies in the decadal $\delta^2H_{LM}$ variability among the four *Fagus sylvatica* trees as noted for F1
(1916 – 1956) as well as for F4 (late 1990's – 2015) (Fig. 3b). Conclusively, despite the complex evolution of
northern Alpine $\delta^2H_{precip}$ changes, the $\delta^2H_{LM}$ chronology and the produced simple linear regression model
suggest that half of the variance (r² = 0.51) can be explained by large-scale annually averaged temperature
changes.

**5. Conclusions**

In this study, we investigated in detail the relationship between inter-annually resolved $\delta^2H_{LM}$ values and (site-
specific) instrumental $\delta^2H_{precip}$ as well as local and large-scale climate data by using nine annually resolved
*Fagus sylvatica* tree-ring series collected at Hohenpeißenberg (Germany). For the common period of overlap
(1916-2015), the coherency among the nine $\delta^2H_{LM}$ tree-ring series is indicated by high inter-series correlations
(Rbar = 0.52). However, some trees show partial deviations in their temporal $\delta^2H_{LM}$ trends, which, however,
seem neither be associated with known stable isotope mechanisms affecting juvenile tree-rings nor related to
specific tree ages in general. Nonetheless, the significant overall inter-series correlation gives confidence to
produce an averaged $\delta^2H_{LM}$ chronology (integrating the nine tree-ring series) subsequently being used to assess
co-variances with $\delta^2H_{precip}$ and climate parameters. Locally, the highest correlations between the $\delta^2H_{LM}$
chronology were generally found with annually averaged values of $\delta^2H_{precip}$ (r = 0.73, p<0.001) and temperature
(r = 0.56, p <0.001) particularly when considering the year prior the latest late wood formation as, for instance,
from previous September to current August. Correlations with temperature further increased when compared to
large-scale temperature anomalies across Western Europe showing r values higher than 0.6 for multiple (largely
connected) areas. Those areas largely overlap with modern moisture source areas of the Hohenpeißenberg study
site indicating that remote meteorological changes primarily control the variance of the $\delta^2H_{LM}$ chronology. As





the $\delta^2H_{LM}$ chronology shows significant correlations with surface air temperatures for largely connected areas across western Europe (these areas also largely overlap over time), we produced a linear regression model between average surface air temperature anomalies of western Europe and the $\delta^2H_{LM}$ chronology (r = 0.71, p<0.1). When comparing instrumental and reconstructed large-scale temperature anomalies, an average absolute

deviation in annual reconstructions of 0.3 °C was found (n = 100). Therefore, $\delta^2H_{LM}$ values of mid-latitudinal tree-ring archives are considered suitable for reconstructions of large-scale mean annual temperature variability and may therefore particularly complement commonly used plant physiological tree-ring proxies (reflecting local temperature variability) yielding an improved paleoclimatic potential of Late Holocene tree-ring archives.

### Data availability

The data used in this publication are available to the community and can be accessed by request to the corresponding author.

### Author contributions

FK and TA designed the study. Financial support acquired by FK. BS and TA carried out field campaigns guided by WT. Tree-ring dating conducted by BS, CH and JE. BS, TA, FK and MG generated stable isotope data. TA,

CH, WT, DS, JS and FK interpreted the data and wrote the paper.

### Competing interests

The authors declare no competing financial interests.

### Acknowledgments

This study was supported by the Deutsche Forschungsgemeinschaft (DFG; KE884/6-2, KE884/6-3, KE884/8-1

and KE884/8-2, SCHO 1274/13-1). We gratefully thank Moritz Schroll and Hendric Glatting for supporting field campaigns.

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



Table 1 – Age coverage of the nine tree-rings of *Fagus Sylvativa*

| Series ID | Pith offset | First year | Last year |
|---|---|---|---|
| F1.1 | 7 | 1912 | 2015 |
| F1.2 | 0 | 1905 | 2015 |
| F1.3 | 7 | 1912 | 2015 |
| F2.1 | 23 | 1858 | 2015 |
| F2.2 | 8 | 1844 | 2015 |
| F3.1 | 7 | 1890 | 2015 |
| F3.2 | 33 | 1916 | 2015 |
| F4.1 | 6 | 1896 | 2015 |
| F4.2 | 26 | 1916 | 2015 |





**Table 2** – Descriptive statistics of the nine $\delta^2H_{LM}$ tree-ring series of the four *Fagus sylvativa* trees

| Series ID | $\delta^2H_{LM}$ [‰ vs. VSMOW] over 1916-2015 | | | | |
|---|---|---|---|---|---|
| | Minimum | Maximum | Difference (Max. − Min.) | Mean with SD (1σ) | Lag-1 |
| F1.1 | −294 | −243 | 51 | −269 ± 11 | 0.645 |
| F1.2 | −288 | −243 | 45 | −266 ± 10 | 0.515 |
| F1.3[a] | −295 | −236 | 59 | −271 ± 11 | 0.408 |
| F2.1 | −270 | −228 | 43 | −250 ± 8 | 0.495 |
| F2.2 | −287 | −224 | 63 | −251 ± 8 | 0.782 |
| F3.1 | −279 | −231 | 48 | −251 ± 8 | 0.473 |
| F3.2 | −293 | −227 | 66 | −252 ± 13 | 0.598 |
| F4.1 | −277 | −231 | 46 | −261 ± 11 | 0.653 |
| F4.2 | −272 | −231 | 41 | −261 ± 15 | 0.718 |

[a]Nine years (1923-1931) are missing due to analytical problems



**Table 3** – Peason's r (with n) between the nine annually resolved $\delta^2H_{LM}$ tree-ring series (all statistically significant at p < 0.001). Color shading indicates r values between radii of the same tree.

| r / n | F1.1 | F1.2 | F1.3 | F2.1 | F2.2 | F3.1 | F3.2 | F4.1 | F4.2 |
|---|---|---|---|---|---|---|---|---|---|
| F1.1 | | 0.65 | 0.67 | 0.52 | 0.65 | 0.66 | 0.37 | 0.43 | 0.49 |
| F1.2 | 104 | | 0.69 | 0.49 | 0.61 | 0.63 | 0.59 | 0.55 | 0.59 |
| F1.3 | 95 | 95 | | 0.59 | 0.51 | 0.52 | 0.51 | 0.56 | 0.47 |
| F2.1 | 104 | 104 | 95 | | 0.49 | 0.40 | 0.40 | 0.42 | 0.37 |
| F2.2 | 104 | 104 | 95 | 158 | | 0.63 | 0.53 | 0.65 | 0.66 |
| F3.1 | 100 | 100 | 91 | 100 | 100 | | 0.59 | 0.51 | 0.54 |
| F3.2 | 100 | 100 | 91 | 100 | 100 | 100 | | 0.56 | 0.59 |
| F4.1 | 100 | 100 | 91 | 100 | 100 | 100 | 100 | | 0.73 |
| F4.2 | 100 | 100 | 91 | 100 | 100 | 100 | 100 | 100 | |





**Table 4 -** Statistics of the linear regression analysis of the four mean $\delta^2H_{LM}$ tree-ring series over the period 1916-1956 (as
shown in Fig. 4). Series are ordered in accordance to reflected growth phase.

| Series | r (p<0.05) | Slope with 95 % CI [‰/year] | Cambial age covered |
|--------|-----------|-----------------------------|---------------------|
| F2 | 0.39 | 0.24 ± 0.18 | 81 - 121 (mature) |
| F3 | 0.63 | 0.34 ± 0.14 | 34 - 74 (intermediate) |
| F4 | 0.50 | 0.31 ± 0.17 | 27 - 67 (intermediate) |
| F1 | 0.67 | 0.49 ± 0.18 | 12 - 52 (juvenile) |





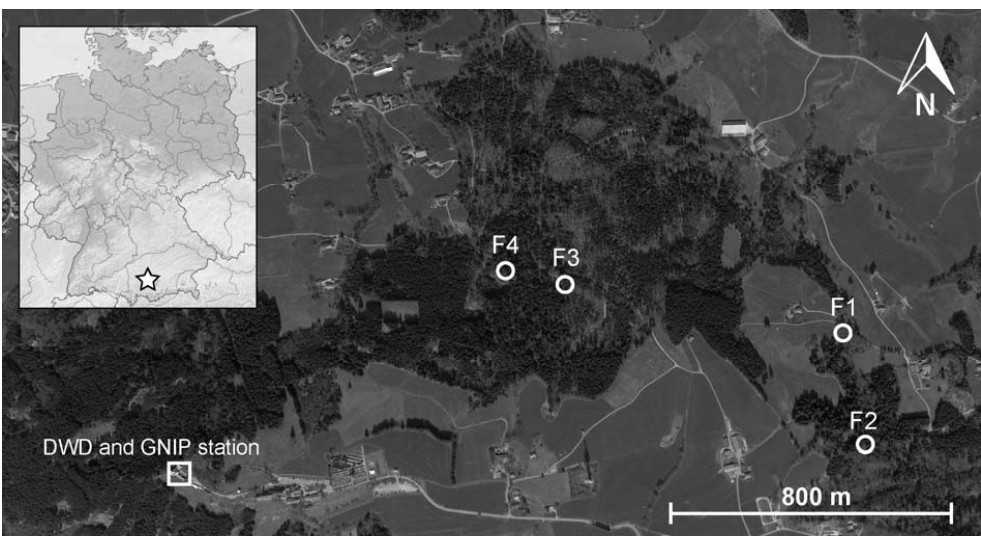

**Fig. 1** – Map of the investigation area. Top left shows the location of the sampling site Hohenpeißenberg (white star) in Germany. Circles in the aerial picture mark the location of the four *Fagus sylvatica* L. specimens (F1-F4) used for tree ring $\delta^2H_{LM}$ measurements which are within a distance of 2 km from the Deutsche Wetterdienst (DWD) and the Global Network of Isotopes in Precipitation (GNIP) stations (square).




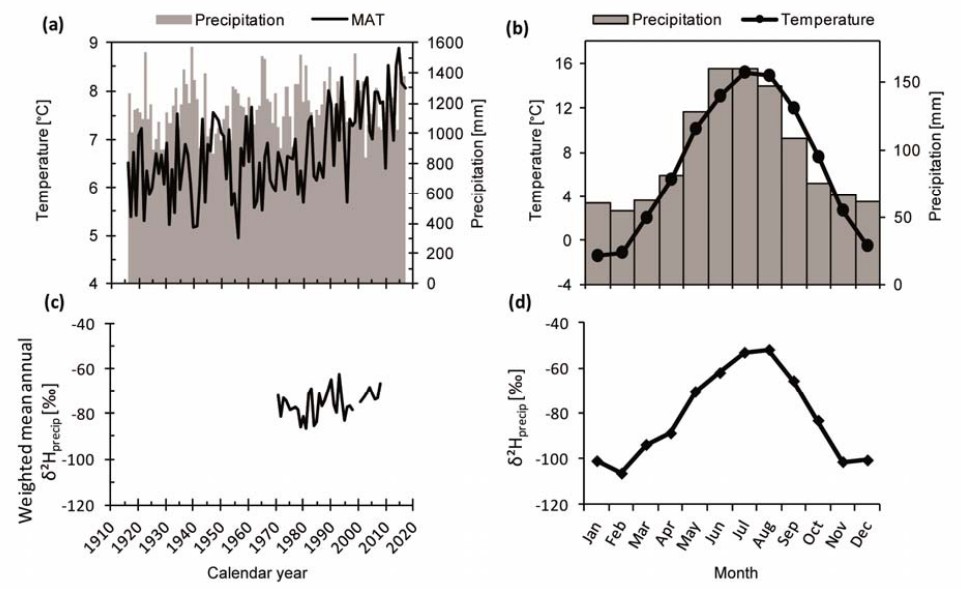

**Fig. 2** – Instrumental climate and $\delta^2H_{precip}$ data from the DWD and GNIP stations at Hohenpeißenberg. **(a)** Mean annual temperature (MAT) and total precipitation since 1916. **(b)** Seasonal temperature and precipitation patterns over the 1916-2015 period. **(c)** Weighted mean annual $\delta^2H_{precip}$ values for the 1971-2008 period (the years 1999 and 2000 are missing due to insufficient data). **(d)** Monthly resolved mean $\delta^2H_{precip}$ values for the 1971-2008 period.



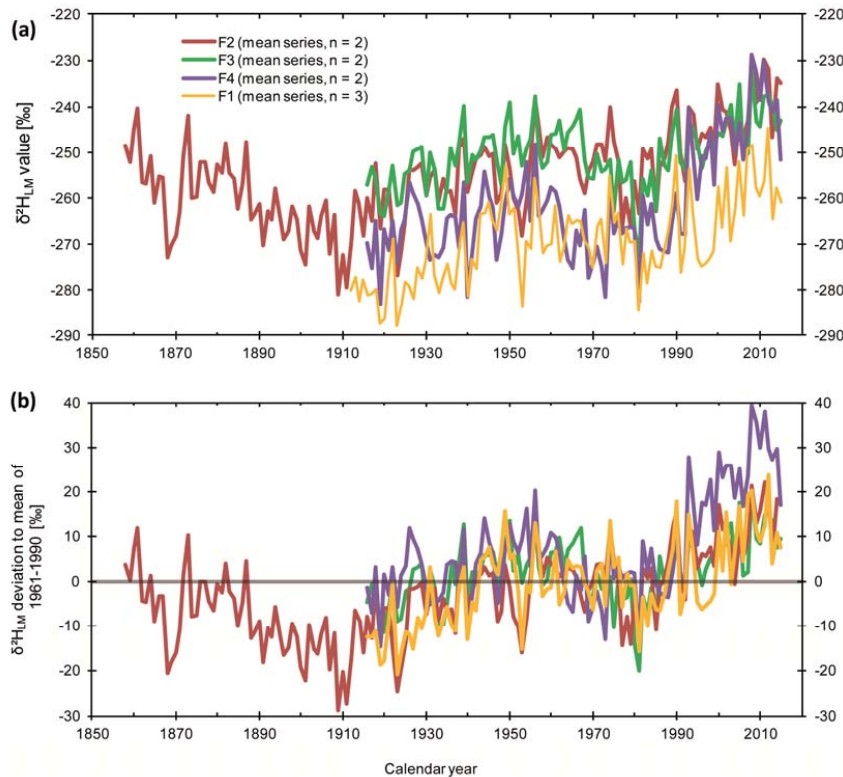

**Fig. 3 – (a)** Annually resolved $\delta^2H_{LM}$ values of the four *Fagus Sylvatica* trees (F1, F2, F3 and F4) collected at Hohenpeißenberg (Germany). Each series reflects the mean $\delta^2H_{LM}$ value of n = 2 or 3. **(b)** Standardized tree-ring series of (a) using the mean $\delta^2H_{LM}$ value of the reference period 1961-1990.

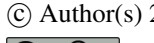



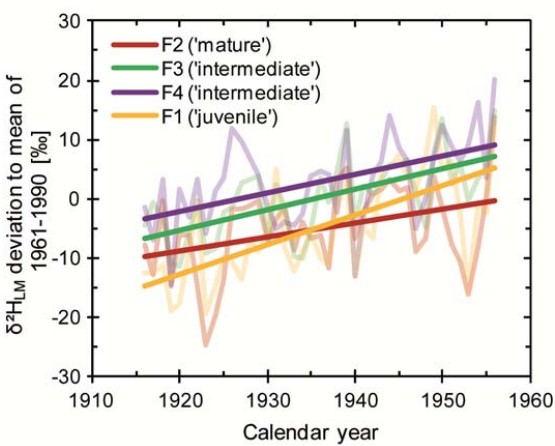


**Fig. 4** – Linear regression analysis of the four mean δ²H$_{LM}$ tree-ring series over the period 1916-1956. During this period, trees differ in age. Statistics of linear fits are given in Table 4.





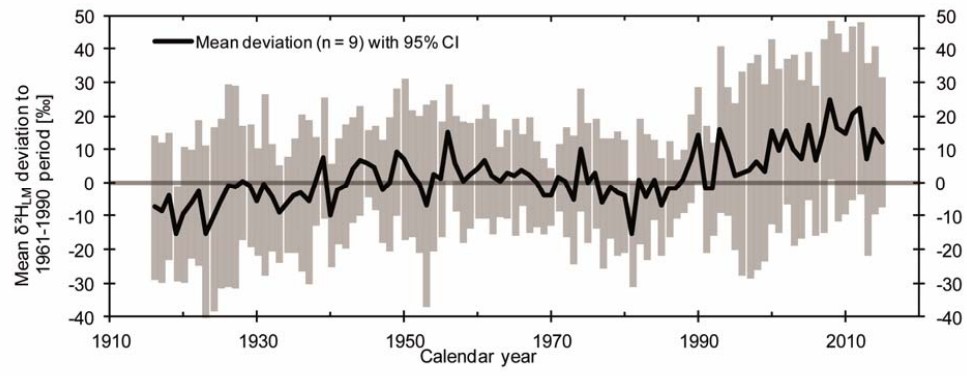

**Fig. 5** – $\delta^2H_{LM}$ chronology for the 1916-2015 common period. Chronology reflects the arithmetic mean of the annual $\delta^2H_{LM}$
deviations with respect to the 1961-1990 period of the nine individual tree-ring series and is shown with 95 % confidence
intervals (grey bars).





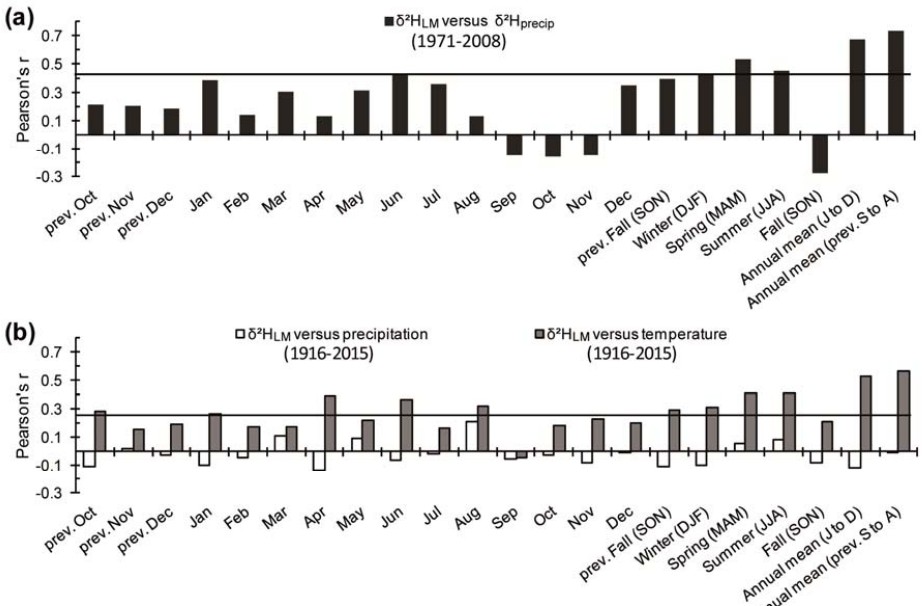

**Fig. 6 -** Pearson's correlations (r) between annually resolved mean δ²H$_{LM}$ chronology (cf. Fig. 5) and δ²H$_{precip}$ from 1971 –
2008 (n = 36) (**a**) as well as with temperature and precipitation from 1916-2015 (n = 100) (**b**) for Hohenpeißenberg
540    (Germany). Horizontal lines mark significance level of 99.9%.





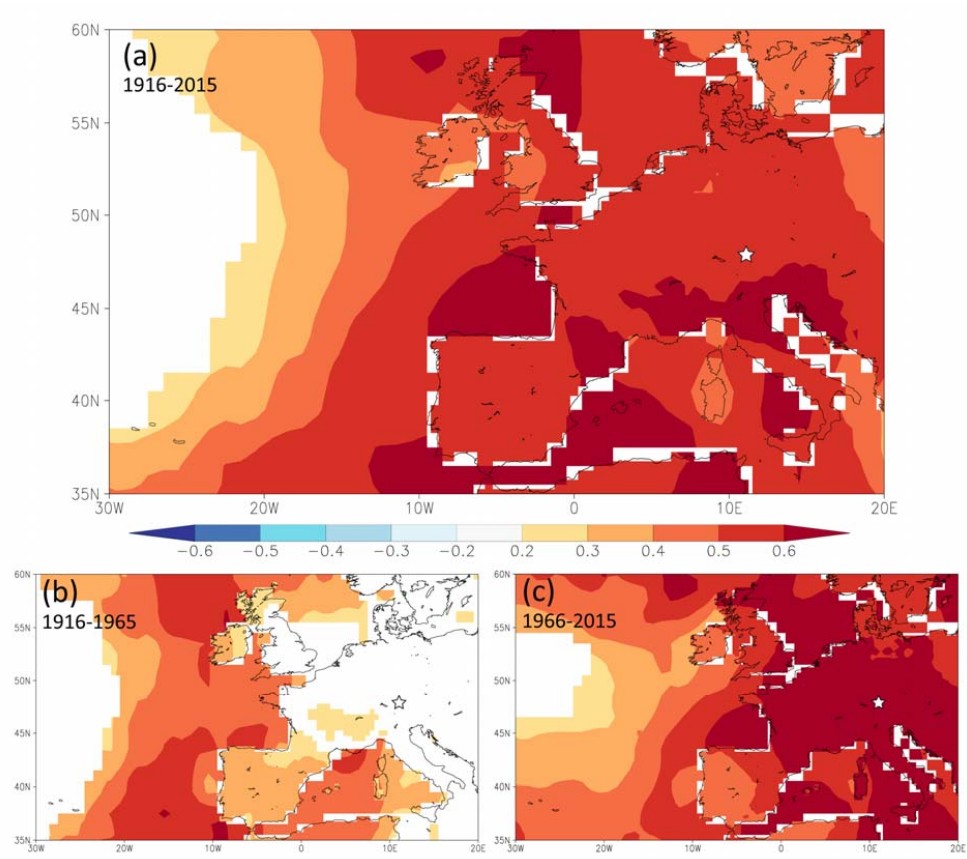

**Fig. 7** – Spatial correlations between inter-annually resolved land and sea air temperature changes (previous September to current August; CRUT4.01 and HadSST1, respectively) and mean $\delta^2H_{LM}$ chronology from Hohenpeißenberg (Germany) for the time period 1916-2015 (**a**) as well as for the time periods 1916-1965 (**b**) and 1966-2015 (**c**). Only correlations with significance $p < 0.1$ are shown. White stars mark the Hohenpeißenberg study site in Germany.





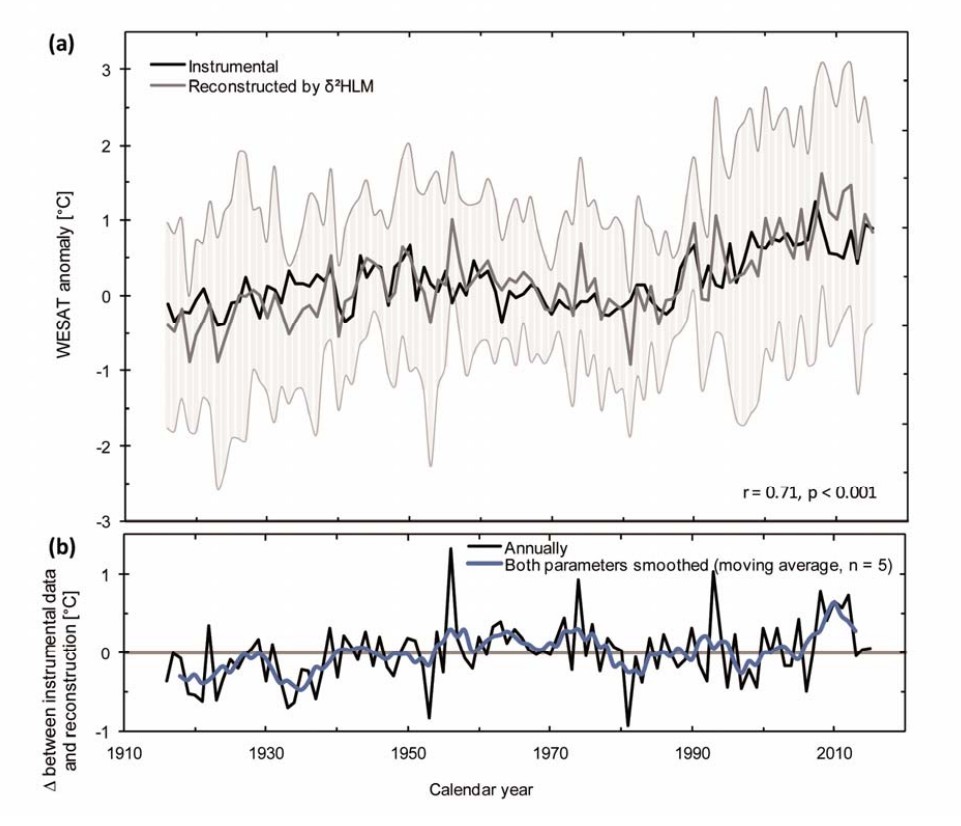

**Fig. 8 – a)** Instrumental and reconstructed WESAT anomaly (using Eq. (1)) for the time period 1916-2015 at annual
resolution. Equation (1) was also used to convert the annually estimated 95% confidence interval into temperature (grey bars)
to account for errors induced by the internal $\delta^2H_{LM}$ variability of the nine tree-ring series **b)** Difference between reconstructed
550     and instrumental WESAT anomaly at annual resolution and with a moving average of 5 of both parameters.