# Peer review of "Annually resolved $\delta^2 H$ tree-ring chronology of the lignin methoxyl groups from Germany reflects averaged Western European surface air temperature changes"

_Climate of the Past, 2019_

## Referee Comment (RC1) · Anonymous Referee #1 · 19 Feb 2019

This paper investigated the potential of using stable hydrogen isotopes ratios of lignin methoxyl groups to reconstruct large-scale temperature anomalies from the year 1916 to 2015. In my view, there are methodological problems. In particular, robustness of the relationship on which the reconstruction is based. The relationship between the tree-ring $\delta 2HLM$ and surface air temperature seems non-linear through time, which discard the possibility of any reconstruction. In addition, I found it difficult to see the originality of the paper and some results are redundant compared to Anhäuser et al 2017 "Stable hydrogen isotope values of lignin methoxyl groups of

four tree species across Germany and their implication for temperature reconstruction".

Please also note the supplement to this comment:
https://www.clim-past-discuss.net/cp-2019-8/cp-2019-8-RC1-supplement.pdf

**Supplement:**

[revised manuscript text omitted]

$\delta^2 H_{LM}$ and mean annual $\delta^2 H_{precip}$ values  observed (implying that $\delta^2 H_{sw}$ values reflects an annual

80    site-specific $\delta^2 H_{precip}$ valu) $\delta^2 H_{sw}$ values were not measured in most of these studies the estimated isotope fractionation was calculated between $\delta^2 H_{precip}$ and $\delta^2 H_{LM}$ (commonly expressed as the apparent isotope fractionation $\varepsilon_{app}$)  is considered to reflect primarily the biosynthetic isotope fractionation () during lignin methoxyl biosynthesis.

Calculations of $\varepsilon_{app}$ showed a broad agreement between coniferous and deciduous tree-species () with $\varepsilon_{app}$

85   values of –213 ± 17 ‰  *Picea abies* (–237 ± 19 ‰; Anhäuser et al., 2017a). At a single site, $\delta^2 H_{LM}$ values showed maximum differences of ≤ 28 ‰ between five trees (within a circumference of 200 m). Hence, the accompanied variability of $\varepsilon_{app}$ (in the range of ± 17 to 19 ‰) was mainly assigned to inter-tree variability and defines  the accuracy to reconstruct 'absolute' $\delta^2 H_{precip}$ values using $\delta^2 H_{LM}$ values. Reconstructions of 'absolute' $\delta^2 H_{precip}$ values have therefore been considered suitable  
[revised manuscript text omitted]

550

---

## Short Comment (SC1) · 4 Mar 2019

**Reply to comment of referee #1**

I want to greatly thank the referee for the constructive and helpful comments, and address here the main concerns raised by the referee. As correctly noted, the relationship between averaged surface air temperatures and the $\delta^2H_{LM}$ chronology indicates inconsistency over time (cf. line 322-328 and Fig. 7b and c). I agree with the referee that this observation reveals the current drawback to quantitatively reconstruct temperature changes using tree-ring $\delta^2H_{LM}$ values and requires in a revised manuscript not only further highlighting but also guidance on how to address this issue in future studies. This will also involve an attenuation of the current paleoclimatic potential of tree-ring $\delta^2H_{LM}$ values. I also agree with referee #1 that the submitted manuscript insufficiently emphasized the originality of our investigation particularly with respect to earlier studies. Anhäuser et al. (2017a,b) quantified and evaluated primarily the spatial $\delta^2H_{precip}$-$\delta^2H_{LM}$ relationship using numerous sampling sites. However, in both studies $\delta^2H_{LM}$ values were exclusively measured from homogenized tree-ring sections of one or two decades (this crucial information is missing in the submitted manuscript). On the contrary, the here submitted manuscript firstly describes the temporal $\delta^2H_{precip}$-$\delta^2H_{LM}$ relationship at a single site and annual resolution. Hence, this approach sets the current study clearly apart from earlier investigations.

Overall, by addressing the raised main concerns of referee #1 in accordance with the above outlined brief respond, the manuscript is considered to notably improve in a revised version. Therein we will further address all comments of the detailed supplemental review of referee #1.

Sincerely, Tobias Anhäuser

---

## Referee Comment (RC2) · Anonymous Referee #2 · 19 Mar 2019

**Review of "Annually resolved $\delta^2$H tree-ring chronology of the lignin methoxyl groups from Germany reflects averaged Western European surface air temperature changes" by Anhäuser et al., submitted to Climate of the Past**

This paper describes a study of lignin $\delta^2$H measurements from an Alpine site in Central Europe. Lignin isotope studies complement existing studies based on cellulose, because they it is formed in the xylem and thus less affected by enriched leaf water. The study explores some of the potential influences of the $\delta^2$H$_{LM}$ that can lead to variation between individual trees at the same site, and attempts an air temperature

reconstruction for Western Europe. Comparison to an observed temperature record appears promising (with uncertainties). The study is generally well written. I have several comments that could lead to a more comprehensive discussion of some of the findings. I suggest acceptance of the manuscript subject to minor revisions.

**Main comments**

1. In the introduction, the deficiencies of cellulose isotope measurements are discussed. It would be very useful to compare your results with any existing isotope record from cellulose in the region.

2. The groundwater influence is now mostly discussed as a hypothetical factor. It seems unlikely that the groundwater stable isotope composition would vary systematically across the sites at a distance of up to 1 km.

3. There are some studies on the isotope composition of runoff, precipitation and water vapour in alpine/subalpine catchments which could enlighten the discussion of the results (Fischer et al. 2017 and references therein, Aemisegger et al., 2014).

4. The soil properties and hydrology could be discussed more explicitly. What is the role of snow melt in the local hydrology? What is the soil type, is there permafrost?

5. The correlation presented in Fig. 7 seems less stable through time than the discussion suggests. The early part is not correlated with the local influences whereas the latter is. Does the same correlation pattern appear for these time slots when you use observed temperature? That way you could reinforce that a climatic signal is picked up, otherwise you would have an indication that the correlation with the isotope measurements is not stable through time.

6. It would be helpful to show some selected correlations between $\delta^2\mathrm{H}_{LM}$ and d2Hprecip that are summarized now in Fig. 6 in detail, such as for the annual mean, spring and summer.

7. I am used to ordering multiple references that go with one sentence be by time, rather than alphabetically.

**References**

Fischer, B M C, H J I van Meerveld, and J Seibert. 2017. "Spatial Variability in the Isotopic Composition of Rainfall in a Small Headwater Catchment and Its Effect on Hydrograph Separation." Journal of Hydrology 547: 755–69. doi:10.1016/j.jhydrol.2017.01.045.

Aemisegger, F, S Pfahl, H Sodemann, I Lehner, S I Seneviratne, and H Wernli. 2014. "Deuterium Excess as a Proxy for Continental Moisture Recycling and Plant Transpiration." Atmospheric Chemistry and Physics 14: 4029–54. doi:10.5194/acp-14-4029-2014.

---

## Author Comment (AC1) · 9 May 2019

**Response letter to referee 2**

**General comment of referee 2**

This paper describes a study of lignin $\delta^2H$ measurements from an Alpine site in Central Europe. Lignin isotope studies complement existing studies based on cellulose, because they it is formed in the xylem and thus less affected by enriched leaf water. The study explores some of the potential influences of the $\delta^2H_{LM}$ that can lead to variation between individual trees at the same site, and attempts an air temperature reconstruction for Western Europe. Comparison to an observed temperature record appears promising (with uncertainties). The study is generally well written. I have several comments that could lead to a more comprehensive discussion of some of the findings. I suggest acceptance of the manuscript subject to minor revisions.

**General response to referee 2**

We appreciate the time and effort Referee #2 has spent on reviewing our manuscript. Below, we address each comment separately and highlighted as follows: *Referee comments* → **Response** → **Changes**. Line referencing refers to the originally submitted manuscript.

**Response to main comments of Reviewer 2**

*1.        In the introduction, the deficiencies of cellulose isotope measurements are discussed. It would be very useful to compare your results with any existing isotope record from cellulose in the region.*

**Response:** We fully agree that a comparison of the stable isotope signatures from both compounds would be useful. However, to the best of our knowledge, there is no tree-ring chronology of cellulose isotope measurements from the region available with a matching age coverage that would allow a meaningful comparison with our $\delta^2H_{LM}$ chronology. We note that high-altitude $\delta^2H_{cellulose}$ or $\delta^{18}O_{cellulose}$ chronologies from the Alps are available in the literature. However, we consider stable isotope chronologies from such sites as not suitable for a comparison with the Hohenpeißenberg $\delta^2H_{LM}$ chronology due to potential additional affects on stable water isotopes, such as the altitude effect. **Changes:** Unless the referee is aware of any suitable cellulose chronology that we could consider for such a comparison, we suggest no changes.

*2.        The groundwater influence is now mostly discussed as a hypothetical factor. It seems unlikely that the groundwater stable isotope composition would vary systematically across the sites at a distance of up to 1 km.*
*3.        There are some studies on the isotope composition of runoff, precipitation and water vapour in alpine/subalpine catchments which could enlighten the discussion of the results (Fischer et al. 2017 and references therein, Aemisegger et al., 2014).*

**Combined response:** We agree that local variabilities in groundwater $\delta^2H$ values are an important control on the $\delta^2H_{precip}$-$\delta^2H_{LM}$ relationship and can - with the help of other studies - be involved more explicitly in our discussion. However, the suggested studies rather focus on the spatial variability of $\delta^2H_{precip}$ values of single or few precipitation events (Aemisegger et al., 2014; Fischer et al., 2017). Since we show strong evidence that $\delta^2H_{LM}$ values rather reflect an annual mean of local $\delta^2H_{precip}$ (cf.

Fig. 6a), we consider it more suitable to include studies that investigated spatial $\delta^2$H variability of soil water. For instance, the recent study of Goldsmith et al. (2019) analyzed soil water $\delta^2$H values across an 1 ha large temperate forest in central Switzerland (720 m.a.s.l.). They further investigated $\delta^2$H values of water extracted from *Fagus sylvatica* branches from the same area and concluded that the trees primarily used water from deeper soil layers (40 – 50 cm), which showed $\delta^2$H differences of up to 21 ‰. Consequently, there is further evidence that the maximum difference in mean $\delta^2$H$_{LM}$ values among our sampled trees (21 ‰) results from systematic $\delta^2$H differences in soil water. **Changes:** We will integrate the findings of Goldsmith et al. (2019) in our discussion of spatial variability of $\delta^2$H$_{precip}$ and $\delta^2$H$_{LM}$ values.

*4.       The soil properties and hydrology could be discussed more explicitly. What is the role of snow melt in the local hydrology? What is the soil type, is there permafrost?*

**Response**: Please see the response above regarding the extended discussion of the hydrology and soil properties. Snow falls almost exclusively in December, January and February at Hohenpeißenberg. The study area consists of eutric cambisols and has no permafrost. Adding this information appears helpful although the main aim of this study was to investigate (for the first time) the relationship between instrumental $\delta^2$H$_{precip}$ and $\delta^2$H$_{LM}$ tree-ring series at annual resolution. The potential influences of varying soil properties or hydrology among the micro-sites on the $\delta^2$H$_{precip}$-$\delta^2$H$_{LM}$ relationship in space and time require a different study design, such as applied by Goldsmith et al. (2019). Even though these are essential aspects for future studies, they are considered to be beyond the scope of this paper. **Changes:** The requested information regarding snow fall, soil type and absence of permafrost will be added to Section '2. Study site and local climate'.

*5.       The correlation presented in Fig. 7 seems less stable through time than the discussion suggests. The early part is not correlated with the local influences whereas the latter is. Does the same correlation pattern appear for these time slots when you use observed temperature? That way you could reinforce that a climatic signal is picked up, otherwise you would have an indication that the correlation with the isotope measurements is not stable through time.*

**Response:** We estimated temporal variance in correlation between local temperatures (using the 'shifted' annual average) and the $\delta^2$H$_{LM}$ chronology for two 50 years periods. This comparison yields significance for the recent period of 1966-2015 (r = 0.64, p<0.01) and insignificance for the earlier period 1916-1965 (r = 0.24; p > 0.1). Hence, the temporal inconsistency in the relationship between local temperature and $\delta^2$H$_{LM}$ agrees with the observed eastward expansion of areas of highest correlation over time (cf. Fig 7b and c). **Changes:** We will add the correlation coefficients mentioned above in Section '4. Results and Discussion'. As mentioned in SC1 to referee #1, we agree that the observed relationship between (large-scale) averaged surface air temperatures and the $\delta^2$H$_{LM}$ chronology indicates inconsistency over time. Even though this current drawback was mentioned and shown in the intital submission (cf. line 322-328 and Fig. 7b and c), a revised manucript requires further highlighting and guidance on how to address this issue in future studies.

*6.       It would be helpful to show some selected correlations between δ2HLM and δ2Hprecip that are summarized now in Fig. 6 in detail, such as for the annual mean, spring and summer.*

**Response:** Agreed. **Changes:** We will produce new figures showing selected $\delta^2H_{LM}$-$\delta^2H_{precip}$ correlations including the annual mean, shifted annual mean, spring and summer. We will include these in the Supplemental section.

*7.        I am used to ordering multiple references that go with one sentence be by time, rather than alphabetically.*

**Response:** The *Climate of the Past* citation style orders multiple references alphabetically. **Changes:** None.

Aemisegger, F., Pfahl, S., Sodemann, H., Lehner, I., Seneviratne, S. I. and Wernli, H.: Deuterium excess as a proxy for continental moisture recycling and plant transpiration, Atmos. Chem. Phys., 14(8), 4029–4054, doi:10.5194/acp-14-4029-2014, 2014.

Fischer, B. M. C., van Meerveld, H. J. (Ilja. and Seibert, J.: Spatial variability in the isotopic composition of rainfall in a small headwater catchment and its effect on hydrograph separation, J. Hydrol., 547, 755–769, doi:10.1016/j.jhydrol.2017.01.045, 2017.

Goldsmith, G. R., Allen, S. T., Braun, S., Engbersen, N., González-Quijano, C. R., Kirchner, J. W. and Siegwolf, R. T. W.: Spatial variation in throughfall, soil, and plant water isotopes in a temperate forest, Ecohydrology, 12(2), doi:10.1002/eco.2059, 2019.

---

## Author Comment (AC2) · 14 May 2019

**Response letter to referee #1**

**General comment of referee #1**

*This paper investigated the potential of using stable hydrogen isotopes ratios of lignin methoxyl groups to reconstruct large-scale temperature anomalies from the year 1916 to 2015. In my view, there are methodological problems. In particular, robustness of the relationship on which the reconstruction is based. The relationship between the tree-ring $\delta^2 H_{LM}$ and surface air temperature seems non-linear through time, which discard the possibility of any reconstruction. In addition, I found it difficult to see the originality of the paper and some results are redundant compared to Anhäuser et al 2017 "Stable hydrogen isotope values of lignin methoxyl groups of four tree species across Germany and their implication for temperature reconstruction".*

**General response**

The authors would like to thank referee #1 for the helpful comments and suggestions. Overall, the changes and modifications resulting from this review are considered to improve the manuscript's quality since our results will be desribed in more detail and the conclusions are considered to be more cautiously drawn. All minor edits regarding rewording (highlighted in the supplemental PDF file) will be considered for change and are not listed below. Below, we address each detailed comment separately and highlighted as follows: *Referee comments* → **Response** → **Changes**. Line referencing refers to the originally submitted manuscript.

**Point to point response to all detailed comments raised by Referee 1**

*The introduction needs to clearly state what is the originality of this work*

**Response:** Agreed. As mentioned in the short comment SC1, earlier studies estimated the *spatial* $\delta^2 H_{precip}$-$\delta^2 H_{LM}$ relationship using numerous sampling sites, whereby solely tree-ring sections were used (homogenized samples covering one to two decades). The here submitted manuscript firstly describes the *temporal* relationship between instrumental $\delta^2 H_{precip}$ and tree-ring $\delta^2 H_{LM}$ values at annual resolution. Hence, this approach sets the current study clearly apart from earlier investigations and is a major step towards the application of tree-ring $\delta^2 H_{LM}$ values for paleoclimatic investigations. **Changes:** We will rephrase line 75ff as follows: "Earlier studies have mainly evaluated and quantified the spatial $\delta^2 H_{precip}$-$\delta^2 H_{LM}$ relationship using numerous sampling sites across continental to global-scale transects (Anhäuser et al., 2017a; Keppler et al., 2007). Therein, $\delta^2 H_{LM}$ analysis was applied on homogenized tree-ring sections covering the most recently collected one or two decades."

*Line 58: tree-ring width can also reflect large scale climate change*

**Response:** Agreed. **Changes:** We will remove the corresponding statements within the parenthesis.

*Line 61: Need references*

**Response:** Agreed. **Changes:** References will be added as follows "e.g., Pauly et al. (2018); Treydte et al. (2006)."

*Line 64: You need to expand a bit here. Are you referring to the cellulose extraction or the signal extraction for the reconstruction?*
*Line 65: You should introduce some nuances here. This was likely true 15 years ago but today no replication might be a reason for rejection.*

**Combined repond:** Agreed to both. We intend to refer here to the chemical cellulose extraction. **Changes:** To clarify both issues, we will combine both sentences as follows: "Furthermore, the chemical extraction procedures prior to stable isotope analyses are time consuming, particularly when aiming for a sufficient replication that allows statistical evaluation of the accompanied $\delta^2$H and $\delta^{18}$O variability among high resolution tree-ring series (McCarroll and Loader, 2004)."

*Line 68: need references*

**Response:** Agreed. **Changes:** References will be added (Anhäuser et al., 2017b, 2017a; Feakins et al., 2013; Keppler et al., 2007; Riechelmann et al., 2017).

*Lines 68-75: But the xylem tissue are formed from the photosynthetic products suggesting that 2H would also be influenced by an evaporative enrichment. You need to add more details and references.*

**Response:** Agreed. **Changes:** We will rephrase the corresponding sentence by mentioning the influence of an evaporative $^2$H enrichment on $\delta^2$H$_{LM}$ as follows: "Lignin is mainly derived from three precursor compounds: p-coumaryl, coniferyl and sinapyl alcohol (Boerjan et al., 2003). Hence, as lignin does not incorporate hydrogen atoms derived from leaf water, $\delta^2$H$_{LM}$ values are not influenced by an evaporative $^2$H enrichment (contrary to cellulose)."

*Line 83: Split the sentence*

**Response:** Agreed. **Changes:** Sentence will be splitted as follows: "As $\delta^2$H$_{sw}$ values were not measured in most of these studies, the estimated isotope fractionation was calculated between $\delta^2$H$_{precip}$ and $\delta^2$H$_{LM}$ (commonly expressed as the apparent isotope fractionation, $\varepsilon_{app}$). It was therefore suggested that $\varepsilon_{app}$ primarily reflects the biosynthetic isotope fractionation ($\varepsilon_{bio}$) during lignin methoxyl biosynthesis."

*Line 84: From this point of the introduction, it is difficult to follow. Do you want to show that those studies have limitations that your study wants to overcome? Sentences clearly stated limitations of those studies may help to see the originality of your work.*

**Response:** Please see our responses above.

*(1) Lines 98-102: Are those details important for your study?*

**Response:** We modified this section. The paragraph was written to highlight that a detailed comparison between $\delta^2$H$_{precip}$ and $\delta^2$H$_{LM}$ at annual resolution has not been conducted so far. The reason that the mentioned studies are not considered suitable for such a comparison primarily results from the lacking site-specific $\delta^2$H$_{precip}$ data (as mentioned in the sentence before) and not

because of their high-alpine origin. Consequently, we consider these details as no longer relevant. **Changes:** Lines 98-102 will be removed from the revised manuscript.

*Lines 103-109: Your objectives are well defined but it is difficult to see the originality of your study within the literature cited before.*

**Response:** OK. **Changes:** Please see responses above.

*Line 103: Is d2HLM already been used for climate reconstruction? If not, this is something you probably want to highlight before in the introduction.*

**Response:** Based on the changes suggested for the introduction, we will highlight that our work represents the first statistical assessment of the *temporal* $\delta^2 H_{precip}$-$\delta^2 H_{LM}$ relationship at inter-annual resolution. Consequently, in the revised manuscript, it will become clear that currently no valid high-resolution climate reconstruction using $\delta^2 H_{LM}$ of tree-ring series is available. **Changes:** Please see responses above regarding changes suggested for the introduction.

*Line 109: to reconstruct Western European surface air temperature changes over the last 100 years.*

**Response:** We assume the referee suggests to add this part at the end of our introduction. We consider this as not appropriate within the introduction as the attempt to reconstruct large-scale averaged temperature changes was initially not intended**. No changes applied**.

*Line 110: A description of the site ecological characteristics is necessary. What is the type of the forest, soil, geology, etc?*

**Response:** Agreed. **Changes:** Descriptions will be added as follows: "The site is located in the northern Alpine foothills consisting of molasse deposits (marlstones and calcareous gravels) and is further characterized by eutric cambisols, no permafrost and deciduous broadleaf forest."

*Line 126: On which basis did you select the 20 trees? What was the criteria to choose 4 trees for the analyses?*

**Response:** Selection of the *Fagus sylvatica* specimen was only based on age (aiming particularly for individuals with a large circumference) to ensure maximum age coverages. Four trees enabled the assessment of inter-tree variability of $\delta^2 H_{LM}$ values and was simply the maximum amount that could be performed considering the analytical costs (the project was financially supported by the German Science Foundation). Furthermore, due to the novelty of this study, no guidance exists regarding the minimum number of trees to be sampled for such an approach. **Changes**: We consider it therefore only necessary to address the first part of the referee´s comment and will add the following sentence: "To assure maximum age coverage, individuals with large circumferences were chosen (2.0 to 4.5 m).".

*Lines 135-137: Those sentences should go earlier with the other dealing with the 4 specimens and cores otherwise it is confusing.*

**Response:** Agreed. **Changes:** Sentences will be moved and integrated accordingly.

*Line 138: F3 and F4 are actually older than what you show on Fig 3. Why did you choose to cut those series? Moreover, this is explaining why F3 and F4 are intermediate in term of juvenile effect and you didn't need to do statistics to show it.*

**Response:** In this paragraph, we mentioned for each tree its longest and shortest tree-ring series (lines 137-140) because each tree provided at least two cores. On the contrary, Fig. 3 displays the mean $\delta^2H_{LM}$ series for each of the four trees (with n ≥ 2), which was shown within the figure and its caption. Thus, each mean $\delta^2H_{LM}$ series starts with the oldest year of the shortest tree-ring series. In the latter part of the comment, we are not sure what statistics the referee is referring to. Tree age was determined using the pith offset and not by a statistical approach. **Changes:** We see no need for changes.

*Line 189: potentially? As of now, you haven't show this link yet.*

**Response:** Agreed. **Changes**: This section will be rephrased to address comments regarding the statistical comparison between $\delta^2H_{precip}$ and $\delta^2H_{LM}$. Please see comment below.

*Line 207ff: The intra-tree variability is not enough discussed. How core series behave within each other? A figure to illustrate the intra-tree variability would be helpful here. How could you be sure that 2 to 3 radii are enough to represent the intra-tree variability? I think you should first present and discuss results of the radii and then inter-tree variability. Otherwise it is confusing.*

*However, it seems that such results have already been discussed in Anhäuser et al., 2017. Sure, not with such a details but you already concluded that "The 'between trees' δ2HLM variability at a single site was ≤ 28 mUr for the four tree species. When investigations from other studies (Feakins et al., 2013) are taken into consideration, we suggest that this variability is mainly induced by heterogeneous source water δ2H values at a single site although some input of noise arising from biosynthetic isotope fractionation should not be ignored. The 'between trees' variability might be reduced by increasing the number of trees for the determination of δ2HLM."*

**Response:** The beginning of the Results and discussion section (line 207ff) intentionally first desribes the general spread of all nine $\delta^2H_{LM}$ series allowing an estimation of differences in 'absolute' values among radii and trees. Subquently, we note a similar inter-tree variability as observed in an earlier study and briefly summarize the conclusion of Anhäuser et al. (2017b). Therefore, we do not understand the criticism regarding repetitive conclusions with respect to earlier studies (please see the latter part of the referee's comment). The repetitive character of this finding is mentioned and the corresponding study cited (cf. line 212-216). In the following paragaph (line 217ff), the consistency of the $\delta^2H_{LM}$ values among radii and trees are assessed using Pearson´s correlation coefficients (Table 3) and Rbar as mentioned in the method´s section. Furthermore, the EPS value of the 1916-2015 period yields 0.91 (please also refer to comment below). An enhanced description of the intra-tree variability over time for each of the four *Fagus sylvatica* trees is considered as not essential since significance has been shown among all $\delta^2H_{LM}$ series. Moreover, the $\delta^2H_{LM}$ chronology (produced later in the MS) comes with a 95% confidence interval accounting for the internal variability among the nine series. We agree, however, that a figure illustrating the intra-tree variability would assisst the statistics presented in Table 3. **Changes:** The EPS value (0.91) will be added in the corresponding sentence and introduced in the section 'Material and methods'. An

additional figure will be produced showing the intra-tree variability of $\delta^2 H_{LM}$ of each of the four trees over time. A preliminary version, considered for the Supplemental section, is shown below.

[Figure]

*Line 210: What does it mean? Is it significant?*

**Response:** The first sentence in Section 4.1 describes the $\delta^2 H_{LM}$ results using simple descriptive statistics allowing the reader to grasp the spread of the stable isotope data. **Changes:** We see no need for changes.

*Lines 210-212: This needs more justification.*

**Response:** This is a description of the results and needs therefore no justification. **Changes:** To clarifiy this issue, we will add both the estimated maximum differences in mean $\delta^2 H_{LM}$ values among trees and the radii in parenthesis and refer to Table 2 in the corresponding sentence.

*Line 220: What does it mean? It is acceptable? What about the EPS value?*

**Response:** Agreed. **Changes:** Please see comment above regarding the EPS value.

*Lines 225-226: Please reformulate*

**Response:** Agreed. **Changes:** The sentence will be changed as follows: "However, a lack of autocorrelation in the $\delta^2H_{precip}$ data may not indicate a lack of autocorrelation for the $\delta^2H_{sw}$ values."

*Line 252: How do you find 0.25 ‰ per year? Earlier you said that the slope of F1 is 0.49‰*

**Response:** The 0.25 ‰ per year reflects the difference between slopes of F1 and F2 (cf. Table 4). **Changes:** We will clarify this issue by adding a corresponding sentence as follows: "Hence, the elevated slope of F1 compared to F2 suggests a juvenile effect of F1 associated with increasing $\delta^2H_{LM}$ values (slope difference between F1 and F2 equals Δ 0.25 ‰ per year)."

*Line 255: I suggest to do a new paragraph here*
*Line 264: This paragraph (lines 264-284) should be moved after the end of the paragraph line 255.*
*Line 284: This (lines 284-288) should be integrated with the paragraph from lines 255-263.*

**Response:** Agreed. **Changes:** Section 4.2 will be restructured as suggested by the referee.

*Line 265: Why referring to arid study when your study is in a temperate region? No other studies exist?*

**Response:** Arid study sites are commonly associated with an evaporative $^2H$ enrichment in upper soil waters. As Hohenpeißenberg is not an arid study site, such an influence may *a priori* be considered as negligible, but cannot be ruled out completely. As non-climatic influences on juvenile $\delta^2H_{LM}$ values have not been investigated yet, we consider it helpful to briefly summarize the usually observed juvenile trends. Subsequently, this allows if and how such influences may affect tree-ring $\delta^2H_{LM}$ values. **No changes applied**.

*Line 289: Here you could wrap up previous results (intra, inter-tree variability, tree age influence,..) in one sentence which will justify to develop the chronology. And then you could start the next sentence by "Therefore, we developed..."*

**Response:** Agreed. **Changes:** In combination with the previous sentence, we will summarize the results as follows: "A better understanding of such occasional temporal trend deviations in $\delta^2H_{LM}$ values may improve sampling strategies. Trees with potential temporal disturbances in the $\delta^2H_{sw}$-$\delta^2H_{LM}$ relationship may justifiably be excluded when producing a mean $\delta^2H_{LM}$ chronology, subsequently leading to an improved overall coherency. Conclusively, however, the $\delta^2H_{LM}$ values of the nine tree-ring series show strong a overall coherency and no major influence related to tree age. Therefore, we developed ... ".

*Lines 298-300: You have to expand here. How April, May, July, August and September are not significant whereas it is the growing season? Moreover, individual winter months are not significant and for some of them the correlation is very weak (prev Oct, Nov, Dec and February). Those months might have a different d2Hprecip because of cold temperature. It is really risky to covered such a long annual period, specially when all processes are not well defined. Combining May to July would make more sense. The highest coefficient correlation shouldn't drive the selected period.*

**Response:** Our $\delta^2H_{LM}$ chronology does not only correlate best with the 'shifted' annual value, but we are also convinced to have provided a plausible hydrological mechanism. In Section 3.3 (line 191-

194), we explained the involvement of annually averaged $\delta^2H_{precip}$ values in our statistical approach. Tree source water can involve precipitation of multiple seasons potentially even integrating events from the previous year (Ehleringer and Dawson, 1992; Feng and Epstein, 1995; Tang et al., 2000). A more recent study by Allen et al. (2019) demonstrated in detail that winter precipitation is not only involved in spring/summer soil water, it may even be the dominant contributor. Consequently, the highest correlation between $\delta^2H_{precip}$ and $\delta^2H_{LM}$ values found for the annually averaged value is a strong indication that the tree source water reflects best an annual integral of the local precipitation. We agree, however, that the relevance of these hydrological considerations should be discussed in more detail.

**Changes:** We will modify the corresponding section '3.3 $\delta^2H_{LM}$ covariance assessment and calibration' as follows: "At humid sites, such as Hohenpeißenberg, soil water drawn during the tree´s growth period rather reflects a temporal integral of multiple precipitation events fallen both during and prior to the growth period, potentially even involving winter precipitation of the previous year (Allen et al., 2019; Ehleringer and Dawson, 1992; Feng and Epstein, 1995; Tang et al., 2000). To estimate the temporal integral of the source water for the *Fagus sylvatica* trees, we calculated Pearson's correlation coefficients between tree-ring $\delta^2H_{LM}$ and $\delta^2H_{precip}$ (GNIP) for the 1971-2008 period using a variety of monthly, seasonal and annual mean values. Moreover, we also included $\delta^2H_{precip}$ signatures of months of the preceding fall as well as 'shifted' annual values, such as previous September to current August."

We will also describe and discuss the resulting correlations coefficients in Section 4.3 in more detail as follows: "The results show that for the 1971-2008 period, correlations between the $\delta^2H_{LM}$ chronology and $\delta^2H_{precip}$ signatures are insignificant when using single months, except for June (r = 0.43, p<0.01; Fig. 6a). Correlations partly increase in magnitude when using averaged $\delta^2H_{precip}$ values of the previous fall, previous winter, spring and summer (r = 0.40; 0.42, 0.53; and 0.45, respectively). Highest r values are, however, found when using annual integrals, such as January until December, with r = 0.66 (p < 0.01; Fig. 6a) or previous September to current August with r = 0.73 (p < 0.01). The observed pattern of the estimated correlation coefficients (r values increase when extending the temporal integral of $\delta^2H_{precip}$ values) clearly indicates that source water of the trees rather reflects an accumulation of multiple months to seasons, which agrees with earlier findings (Allen et al., 2019; Ehleringer and Dawson, 1992; Feng and Epstein, 1995; Tang et al., 2000). Furthermore, tree-ring growth of *Fagus Sylvatica* typically ceases between August (Čufar et al., 2008; Michelot et al., 2012) and September (Kraus et al., 2016). Precipitation falling after the late wood formation can therefore only contribute to the tree source water of the following year. Hence, the $\delta^2H_{LM}$ chronology does not only correlate best with the 'shifted' annual values (previous September to current August), this time period also agrees with considerations in source water accumulation and tree-ring growth."

*Lines 309-311: Yes you found the highest correlation within the year but when looking at month resolution, only April, June and August are highly significant (prev Oct, January are weakly correlated even if significant). This is also surprising that fall doesn't show higher correlation as most of the lignin is formed at the end of growing season. I would have expected to find the highest correlation during fall. How could you explained those discrepancies? Moreover, what are the links between d2Hprecip, d2Hlm and temperature?*

**Response:** We disagree with the referee here. Highest correlations between $\delta^2 H_{LM}$ values and temperature using annual time periods are considered as plausible since a strong indication is provided that the $\delta^2 H_{LM}$ values also reflect an annual integral of $\delta^2 H_{precip}$ (please see response above). Moreover, insignificant correlations for the fall period (September, October, November) are in agreement with *Fagus sylvatica* tree-ring growth. As now embedded in our manuscript (please see response above), tree-ring growth of *Fagus sylvatica* typically ceases between August (Čufar et al., 2008; Michelot et al., 2012) and September (Kraus et al., 2016). The $\delta^2 H_{LM}$ values are mainly the result of a constant fractionation of the $\delta^2 H_{precip}$ values (cf. line 84ff). Consequently, a correlation between $\delta^2 H_{LM}$ and temperature rather mimics a correlation between $\delta^2 H_{precip}$ and temperature. **No changes applied**.

*Lines 315-318: Before doing any estimation, you must address the processes behind your correlation.*

**Response:** Please refer to the responses above.

*Lines 327-328: This result suggests that the relationship between d2HLM and temperature is not linear within time and unfortunately counteracts any possibilities of reconstruction.*

**Response:** As already mentioned in the short comment SC1: 'The relationship between averaged surface air temperatures and the $\delta^2 H_{LM}$ chronology indicates inconsistency over time (cf. line 322-328 and Fig. 7b and c). I agree with the referee that this observation reveals the current drawback to quantitatively reconstruct temperature changes using tree-ring $\delta^2 H_{LM}$ values. **Changes:** The revised manuscript will not only highlight the aforementioned current drawbacks but also provide guidance on how to address this issue in future studies. This will also involve an attenuation of the current paleoclimatic potential of tree-ring $\delta^2 H_{LM}$ values throughout the manuscript.

*Lines 330-332: Split the sentence*

**Response:** Agreed. **Changes:** Will be splitted as follows: "Besides changes in local temperature, $\delta^2 H_{precip}$ values at the site of precipitation are controlled by a number of large-scale and remote hydro-climatic influences. This includes changing meteorological conditions in the moisture source area and meridional atmospheric transport (in particular, modified water volume loss, but also potential mixing with different air masses) (Araguás-Araguás et al., 2000; Dansgaard, 1964; Rozanski et al., 1993)".

*Line 348ff: As the relationship between d2Hlm and surface air temperature seems non linear (Fig 7b and c), I don't see how the reconstruction could be robust. The reason why the reconstruction seems valid it is likely because of the high relationship within the entire period driving by the recent period (1966-2015). To test this relation I would suggest to perform a running correlation through the 1916-2015 period. I suspect that the correlation will be unsignificant within the oldest period and become significant at one time. Moreover, to assess the quality of the model, I suggest to use a 2-fold cross-validation method and compare the RMSE, RE and CE between the 2 period (see Briffa et al. 1988, Cook et al 1999, and Naulier et al 2015). Again, you may find that the statistics aren't significant.*

**Response:** The temporal inconsistency between the $\delta^2 H_{LM}$ values and WESAT anomalies is already indicated in Figs. 7b and c and will be further internalized throughout the manuscript. Thus, we

consider the detailed statistical approach by the referee as not required. **Changes**: Please see response above.

*Lines 353-354: Is it a different set of data than the one used previously in section 4.3? If not, I don't see how you reach 0.71 whereas looking at the map (Fig 7a) the correlation seems between 0.5 and 0.6 around your study site.*

**Response:** For both correlations, we used the same instrumental data set (HadCRUT4.6). However, Fig. 7a shows correlations between $\delta^2H_{LM}$ values and temperature variability for numerous sites across Europe (displayed as 'spatial correlations'), whereby WESAT reflects the average temperature anomaly for the whole area (cf. line 351-352). **Changes:** We will add a sentence to clarify this difference: "The model reveals a significant correlation (r = 0.71, p < 0.001, DW = 1.6) with an elevated r value when compared to any of the observed correlations shown in Fig. 7a".

Allen, S. T., Kirchner, J. W., Braun, S., Siegwolf, R. T. W. and Goldsmith, G. R.: Seasonal origins of soil water used by trees, Hydrol. Earth Syst. Sci., 23(2), 1199–1210, doi:10.5194/hess-23-1199-2019, 2019.

Anhäuser, T., Greule, M., Polag, D., Bowen, G. J. and Keppler, F.: Mean annual temperatures of mid-latitude regions derived from δ2H values of wood lignin methoxyl groups and its implications for paleoclimate studies, Sci. Total Environ., 574, 1276–1282, doi:10.1016/j.scitotenv.2016.07.189, 2017a.

Anhäuser, T., Greule, M. and Keppler, F.: Stable hydrogen isotope values of lignin methoxyl groups of four tree species across Germany and their implication for temperature reconstruction, Sci. Total Environ., 579, 263–271, doi:10.1016/j.scitotenv.2016.11.109, 2017b.

Araguás-Araguás, L., Froehlich, K. and Rozanski, K.: Deuterium and oxygen-18 isotope composition of precipitation and atmospheric moisture, in Hydrological Processes, vol. 14, pp. 1341–1355., 2000.

Boerjan, W., Ralph, J. and Baucher, M.: Lignin biosynthesis, Annu. Rev. Plant Biol., 54(1), 519–546, doi:10.1146/annurev.arplant.54.031902.134938, 2003.

Čufar, K., Prislan, P., De Luis, M. and Gričar, J.: Tree-ring variation, wood formation and phenology of beech (Fagus sylvatica) from a representative site in Slovenia, SE Central Europe, Trees - Struct. Funct., 22(6), 749–758, doi:10.1007/s00468-008-0235-6, 2008.

Dansgaard, W.: Stable isotopes in precipitation, Tellus, 16(4), 436–468, doi:10.3402/tellusa.v16i4.8993, 1964.

Ehleringer, J. R. and Dawson, T. E.: Water uptake by plants: perspectives from stable isotope composition, Plant. Cell Environ., 15(9), 1073–1082, doi:10.1111/j.1365-3040.1992.tb01657.x, 1992.

Feakins, S. J., Ellsworth, P. V. and Sternberg, L. da S. L.: Lignin methoxyl hydrogen isotope ratios in a coastal ecosystem, Geochim. Cosmochim. Acta, 121, 54–66, doi:10.1016/j.gca.2013.07.012, 2013.

Feng, X. and Epstein, S.: Climatic temperature records in δD data from tree rings, Geochim. Cosmochim. Acta, 59(14), 3029–3037, doi:10.1016/0016-7037(95)00192-1, 1995.

Keppler, F., Harper, D. B., Kalin, R. M., Meier-Augenstein, W., Farmer, N., Davis, S., Schmidt, H.-L., Brown, D. M. and Hamilton, J. T. G.: Stable hydrogen isotope ratios of lignin methoxyl groups as a paleoclimate proxy and constraint of the geographical origin of wood, New Phytol., 176(3), 600–609, doi:10.1111/j.1469-8137.2007.02213.x, 2007.

Kraus, C., Zang, C. and Menzel, A.: Elevational response in leaf and xylem phenology reveals different prolongation of growing period of common beech and Norway spruce under warming conditions in the Bavarian Alps, Eur. J. For. Res., 135(6), 1011–1023, doi:10.1007/s10342-016-0990-7, 2016.

McCarroll, D. and Loader, N. J.: Stable isotopes in tree rings, Quat. Sci. Rev., 23(7–8), 771–801, doi:10.1016/j.quascirev.2003.06.017, 2004.

Michelot, A., Simard, S., Rathgeber, C., Dufrêne, E. and Damesin, C.: Comparing the intra-annual wood formation of three European species (Fagus sylvatica, Quercus petraea and Pinus sylvestris) as related to leaf phenology and non-structural carbohydrate dynamics, Tree Physiol., 32(8), 1033–1045, doi:10.1093/treephys/tps052, 2012.

Pauly, M., Helle, G., Miramont, C., Büntgen, U., Treydte, K., Reinig, F., Guibal, F., Sivan, O., Heinrich, I., Riedel, F., Kromer, B., Balanzategui, D., Wacker, L., Sookdeo, A. and Brauer, A.: Subfossil trees suggest enhanced Mediterranean hydroclimate variability at the onset of the Younger Dryas, Sci. Rep., 8(1), 1–8, doi:10.1038/s41598-018-32251-2, 2018.

Riechelmann, D. F. C., Greule, M., Siegwolf, R. T. W., Anhäuser, T., Esper, J. and Keppler, F.: Warm season precipitation signal in δ 2 H values of wood lignin methoxyl groups from high elevation larch trees in Switzerland, Rapid Commun. Mass Spectrom., 31(19), 1589–1598, doi:10.1002/rcm.7938, 2017.

Rozanski, K., Araguás-Araguás, L. and Gonfiantini, R.: Isotopic Patterns in Modern Global Precipitation, pp. 1–36., 1993.

Tang, K., Feng, X. and Ettl, G. J.: The variations in δD of tree rings and the implications for climatic reconstruction, Geochim. Cosmochim. Acta, 64(10), 1663–1673, doi:10.1016/S0016-7037(00)00348-3, 2000.

Treydte, K. S., Schleser, G. H., Helle, G., Frank, D. C., Winiger, M., Haug, G. H. and Esper, J.: The twentieth century was the wettest period in northern Pakistan over the past millennium, Nature, 440(7088), 1179–1182, doi:10.1038/nature04743, 2006.